# CURE-OOD: Benchmarking Out-of-Distribution Detection for Survival Prediction

## Abstract

"How long can I live and remain free of cancer?" is often the first question a patient asks after receiving a cancer diagnosis and treatment. Accurate survival prediction helps alleviate psychological distress and supports risk stratification and personalized treatment planning. Recent survival prediction frameworks have shown strong performance using computed tomography (CT) images. However, variations in imaging acquisition introduce out-of-distribution (OOD) samples caused by covariate shifts that undermine model reliability. Despite this challenge, to our knowledge, no existing benchmark systematically studies OOD detection in cancer survival prediction. To address this gap, we introduce the Cancer sURvival bEnchmark for OOD Detection (CURE-OOD), the first benchmark for systematically evaluating OOD detection in survival prediction under controlled acquisition-induced distribution shifts. CURE-OOD defines scanner-parameter-based training, in-distribution (ID), and OOD test splits across four survival prediction tasks. Our experiments show that covariate shifts notably reduce survival prediction performance. They also show that mainstream classification-oriented OOD detectors can fail when directly adapted to survival prediction. Finally, we include HazRes as a simple survival-aware reference baseline for OOD detection. CURE-OOD enables systematic analysis of how distribution shifts affect both downstream survival performance and OOD detectability.

## 1 Introduction

Accurate survival prediction is critical in cancer prognosis, as it helps clinicians assess patient risk and make informed, personalized treatment decisions. In head and neck cancer, radiation therapy is typically guided by stage-dependent risk stratification protocols, where patients within the same stage receive identical dose prescriptions (Pan et al., 2016; Caudell et al., 2017). Yet patients with similar staging or tumor characteristics can experience markedly different outcomes, ranging from long-term recurrence-free survival to early disease progression or treatment-related toxicity (Ang et al., 2010; Kawecki & Krajewski, 2014). This heterogeneity motivates reliable, individualized survival prediction models that complement traditional staging systems and guide treatment planning (Beesley et al., 2019; Kang et al., 2015; Amin et al., 2017).

Recent advances in deep learning have enabled survival modeling from medical images. Typically, CNNs or Vision Transformers (ViTs) extract imaging features that are passed to survival prediction frameworks (Chen et al., 2024b; Saeed et al., 2024; Chen et al., 2024a). A representative formulation is multi-task logistic regression (MTLR), which reformulates survival analysis into ordered binary subtasks across discrete time intervals to jointly learn a survival distribution (Zhang & Yang, 2021; Yu et al., 2011; Kvamme et al., 2019; Nagpal et al., 2021). In real-world practice, CT scans are often collected over long periods and across hospitals or imaging centers. During this process, scanner hardware and acquisition protocols may be updated or replaced (Welch et al., 2023), while cross-center variation further introduces differences in scanner models and acquisition settings (Guan & Liu, 2021), as illustrated in Fig. 1. This variability changes image appearance and induces covariate shifts while the clinical prediction task remains fixed, creating the deployment gap shown in Fig. 2. In image classification, segmentation, and related medical imaging tasks, such shifts have been shown to reduce model reliability (Gutbrod et al., 2025; Liu et al., 2025; Baek et al., 2024; Zhao

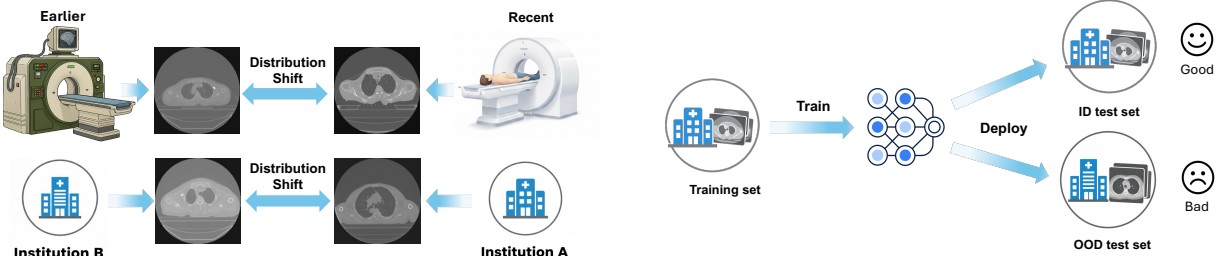

**Figure 1:** CT data distribution shifts caused by cross-institution variability and scanner changes over time.

**Figure 2:** A model that performs well on in-distribution (ID) test data may degrade on out-of-distribution (OOD) test data.

et al., 2025; 2026). However, in CT-based cancer survival prediction, the detectability of acquisition-induced covariate shifts and their downstream impact on survival predictions remain underexplored.

Existing OOD benchmarks and detection methods mainly target classification, where detectors often rely on class-confidence or logit-based signals. Survival prediction differs from classification because its outputs are time-indexed risk or survival distributions rather than class confidence. Therefore, whether classification-oriented OOD detectors remain effective for survival prediction under acquisition-induced shifts has not been systematically studied.

To address these gaps, we introduce CURE-OOD, a benchmark for OOD detection in cancer survival prediction. Table 1 summarizes key differences between CURE-OOD and existing OOD benchmarks. Following standard OOD terminology, ID data are drawn from the same distribution as the training set, whereas OOD data shift from that distribution. We build CURE-OOD as a controlled evaluation protocol with non-overlapping training, ID test, and OOD test splits. These splits are defined by four acquisition parameters, including pixel spacing, exposure time, slice thickness, and X-ray tube current. This protocol enables systematic and reproducible evaluation of two questions. First, whether acquisition-induced covariate shifts degrade survival prediction. Second, whether existing OOD detection methods can reliably identify shifted samples in survival modeling. On CURE-OOD, we find that these shifts reduce survival prediction performance and that naively adapted classification-oriented OOD detectors become unreliable under survival formulations, sometimes even assigning more OOD-like scores to ID samples. We further include HazRes as a simple survival-aware reference baseline that uses predicted hazard as the detection signal.

Our main contributions are summarized as follows:

- We present CURE-OOD, to our knowledge, the first benchmark focused specifically on acquisition-induced OOD detection for cancer survival prediction from real-world clinical CT data, extending medical imaging OOD evaluation beyond classification and segmentation.

- We systematically verify the degradation of survival prediction performance under acquisition-induced covariate shifts and benchmark representative OOD detectors in this setting.

- We introduce HazRes, a simple survival-aware OOD detection baseline based on predicted hazard, and analyze the limitations of existing classification-oriented detectors for survival prediction.

CURE-OOD facilitates more rigorous evaluation of OOD detection in survival prediction and supports the development of survival-aware methods under acquisition-induced shifts.

## 2 Related work

**Survival prediction in medical imaging.** Modern survival modeling has evolved from classical statistics to deep learning. Early work relied on the Cox proportional hazards model (Cox, 1972) and Aalen regression models (Aalen, 1989), which are powerful for statistical inference but limited for individualized prediction

and rely on the proportional hazards assumption. DeepSurv (Katzman et al., 2018) and related deep-Cox approaches (Martinussen & Scheike, 2006; Ching et al., 2018; Nagpal et al., 2021; Kvamme et al., 2019) extend these hazard-based formulations while using neural networks for representation learning. In contrast, MTLR (Yu et al., 2011) discretizes time and learns a sequence of dependent logistic classifiers, directly modeling the survival function and naturally handling censored data. This reformulation makes survival prediction operationally similar to classification across time intervals. In imaging-based survival prediction, MTLR is often used as the survival head on top of learned visual features to predict patient-specific risk or hazard over time (Wiegrebe et al., 2024).

**Distribution shift, robustness, and uncertainty in survival analysis.** Distribution shift is a long-standing challenge in medical survival prediction. Medical imaging datasets are often collected over many years and across multiple institutions, during which scanner hardware, acquisition protocols, and reconstruction algorithms may change substantially (Fortin et al., 2018; Mackin et al., 2015). Such variations alter image appearance and quantitative features, producing covariate shifts that can degrade the performance of downstream survival models (Zech et al., 2018). Similar concerns arise more broadly from differences in patient populations, clinical practices, and outcome distributions across sites and time periods.

To improve reliability under these shifts, prior work has explored robust survival learning, domain generalization, and uncertainty quantification. Stable and robust Cox formulations have been proposed to improve performance under changing covariate distributions (Wen et al., 2014; Fan et al., 2024), while domain-generalization studies evaluate whether survival models trained in one clinical setting transfer to others (Pfisterer et al., 2022). Other work considers label shift in survival analysis (Zong et al., 2025) and conformal prediction under covariate shift (Shin et al., 2025). More broadly, conformalized survival analysis provides distribution-free coverage guarantees for survival quantiles and prediction intervals through post-hoc calibration (Candes et al., 2023; Gui et al., 2024). These studies target robustness, calibration, and valid coverage under censoring, but they do not directly evaluate whether acquisition-shifted test samples can be detected.

**OOD detection in medical imaging and survival prediction.** OOD detection aims to identify inputs that deviate from the training distribution before their predictions are trusted. In medical imaging, prior benchmarks have evaluated OOD detection across modalities and tasks (Cao et al., 2020), while Open-MIBOOD (Gutbrod et al., 2025) extends standardized OOD evaluation to medical image classification. Nevertheless, most existing OOD detectors are designed for classification models, whose outputs are class logits or softmax probabilities. *Output-based* scores leverage classifier confidence, including maximum softmax probability (MSP) (Hendrycks & Gimpel, 2016), ODIN (Liang et al., 2017), maximum logit scoring (MLS) (Hendrycks et al., 2019), energy-based scoring (Liu et al., 2020), and generalized-entropy scoring (GEN) (Liu et al., 2023). *Activation- and weight-shaping* methods modify penultimate activations or classifier weights before scoring, such as ASH (Djurisic et al., 2022), DICE (Sun & Li, 2022), and SCALE (Xu et al., 2024). *Representation-based* methods score samples in feature space. NNGuide (Park et al., 2023) follows this paradigm by leveraging nearest-neighbor information in the learned representation space. Another uncertainty-based approach is Monte Carlo dropout (Dropout), which estimates predictive uncertainty through stochastic forward passes (Gal & Ghahramani, 2016). Related work has also extended OOD detection to segmentation, where the goal is often pixel-level anomaly localization (Zhao et al., 2025; 2024; Vojíř et al., 2024; Shoeb et al., 2024).

OOD detection remains underexplored for survival prediction. Prior survival-shift studies mainly focus on robust modeling, domain generalization, calibration, or coverage under distribution shift. In contrast, OOD detection asks whether shifted test samples can be identified before their predictions are trusted. This is challenging because survival models produce time-indexed risk or survival estimates under censoring-aware objectives, while most existing OOD scores are built around classification-style confidence outputs.

**Table 1:** Comparison with existing OOD benchmarks. While most existing benchmarks focus on natural image classification under semantic or synthetic shifts, CURE-OOD addresses acquisition-induced covariate shifts in clinical imaging for survival prediction.

| Benchmark | Med. | Primary Task | Shift Type | Source |
|---|---|---|---|---|
| OpenOOD (Yang et al., 2022) | ✗ | Classification | Semantic | Mixed |
| OpenMIBOOD (Gutbrod et al., 2025) | ✓ | Classif. / Segmen. | Mixed | Real |
| ImageNet-ES (Baek et al., 2024) | ✗ | Classification | Covariate | Real |
| ImageNet-C/ CIFAR-C (Hendrycks & Dietterich, 2019) | ✗ | Classification | Covariate | Synthetic |
| ImageNet-A/-O/-R (Hendrycks et al., 2021) | ✗ | Classification | Semantic | Real |
| WILDS (Koh et al., 2021) | Part. | Classif. / Regress. | Covariate | Real |
| MIDOG (Aubreville et al., 2023) | ✓ | Detection | Covariate | Real |
| PhaKIR (Rueckert, 2024) | ✓ | Detect. / Classif. | Covariate | Real |
| OASIS-3 (LaMontagne et al., 2019) | ✓ | Classification | Covariate | Real |
| MedMNIST v2 (Yang et al., 2023) | ✓ | Classification | Mixed | Mixed |
| BREEDS (Santurkar et al., 2020) | ✗ | Classification | Subpopulation | Real |
| **CURE-OOD** | ✓ | Risk modeling | Covariate | Real |

# 3 Background

## 3.1 Survival Analysis

Survival prediction aims to estimate the likelihood of a clinical event, such as cancer recurrence or death, occurring over time. Unlike standard classification tasks that predict discrete labels, survival models estimate a time-dependent probability distribution describing event risk across time intervals. In medical imaging, such models leverage features extracted from CT scans to capture patterns associated with disease progression. Among existing formulations, multi-task logistic regression (MTLR) (Yu et al., 2011) is particularly suitable for censored survival prediction because it represents the task as a sequence of dependent binary classification tasks across discretized time intervals.

Let $\{t_1, \ldots, t_m\}$ denote $m$ time intervals and $\mathbf{x}$ the features extracted by the visual backbone. MTLR parameterizes the survival probability at each interval as $P_{\theta_i}(T \geq t_i \mid \mathbf{x}) = (1 + \exp(\boldsymbol{\theta}_i^\top \mathbf{x} + b_i))^{-1}$, and enforces temporal consistency through the joint likelihood

$$P_\Theta(Y = y \mid \mathbf{x}) = \frac{\exp\left(\sum_{i=1}^m y_i(\boldsymbol{\theta}_i^\top \mathbf{x} + b_i)\right)}{\sum_{k=0}^m \exp(f_\Theta(\mathbf{x}, k))}, \quad f_\Theta(\mathbf{x}, k) = \sum_{i=k+1}^m (\boldsymbol{\theta}_i^\top \mathbf{x} + b_i). \tag{1}$$

where the resulting logits $f_\Theta(\mathbf{x}, k)$ summarize interval-wise risk over time. Model parameters are learned by minimizing the negative log-likelihood of this distribution.

From the predicted distribution, the event probability is $p_\Theta(\mathbf{x}, t_i) = P_\Theta(T = t_i \mid \mathbf{x})$, the survival probability is $G_\Theta(\mathbf{x}, t_i) = P_\Theta(T \geq t_i \mid \mathbf{x})$, and the discrete hazard is

$$h(t_i \mid \mathbf{x}) = \frac{P_\Theta(T = t_i \mid \mathbf{x})}{P_\Theta(T \geq t_i \mid \mathbf{x})} = \frac{p_\Theta(\mathbf{x}, t_i)}{G_\Theta(\mathbf{x}, t_i)}. \tag{2}$$

## 3.2 Medical Image Acquisition for Survival Analysis

CT appearance is determined by both acquisition and reconstruction settings. Acquisition parameters directly affect image resolution, contrast, and noise characteristics. Variations in these settings, caused by scanner differences or protocol changes across sites and time, lead to systematic changes in image appearance, introducing covariate shifts. Therefore, we can leverage these parameters to define structured distribution shifts. We leverage this property to construct controlled ID and OOD splits, as detailed in Sec. 4.2.

**Table 2:** Dataset partitioning based on acquisition parameters for CURE-OOD. The ID and OOD ranges define the in-distribution and shifted domains for each parameter. Train denotes the number of training cases, while ID Test and OOD Test denote the numbers of test cases sampled from the corresponding domains.

| Parameter | Train | ID Test | OOD Test | ID Range | OOD Range |
|---|---|---|---|---|---|
| Pixel spacing (mm) | 1462 | 100 | 100 | 0.976–1.172 | 0.805–0.969 |
| Exposure time (ms) | 1462 | 100 | 100 | 1000 | 1831–2289 |
| Slice thickness (mm) | 1462 | 100 | 100 | 2 | 2.5 |
| X-ray tube current | 1462 | 100 | 100 | 300 | 305–490 |

## 4 CURE-OOD Benchmark

### 4.1 Data and Preparation

To construct CURE-OOD with controlled acquisition-induced shifts, we require a CT cohort that (i) provides scanner metadata for partitioning and (ii) supports standardized survival endpoints. We therefore construct CURE-OOD by curating and partitioning the RADCURE cohort (Welch et al., 2024), which provides head and neck cancer radiotherapy planning CT scans collected between 2005 and 2017 using scanners from multiple manufacturers. Acquisition parameters are recorded in the DICOM metadata, including KVP, pixel spacing, exposure time, slice thickness, and X-ray tube current. Since all scans were acquired with the same KVP setting, we exclude KVP from the partitioning design. After filtering cases with incomplete CT volumes or missing primary GTVp masks, we retain 2,366 patients for benchmark construction.

Using this filtered cohort, we consider four survival prediction tasks: overall survival (OS), local failure-free survival (LFFS), regional failure-free survival (RFFS), and distant failure-free survival (DFFS). The corresponding numbers of events are 792, 300, 134, and 293, respectively, and we treat each task independently during training and evaluation.

### 4.2 Key Acquisition Parameters

We focus on four CT acquisition parameters that directly influence image appearance and feature distributions: pixel spacing, exposure time, slice thickness, and X-ray tube current. Together, they control effective resolution and noise, leading to realistic acquisition-induced shifts across sites and time (Fig. 3).

**Pixel Spacing** sets the spatial sampling resolution. Smaller spacing preserves finer anatomical details but can amplify noise, while larger spacing smooths textures and yields a wider field of view.

**Exposure Time** controls photon accumulation at the detector. Longer exposure generally improves signal-to-noise ratio and contrast, whereas shorter exposure increases noise and may obscure subtle structures.

**Slice Thickness** determines the degree of through-plane averaging. Thicker slices reduce random noise but blur edges and fine structures, while thinner slices preserve sharper boundaries at the cost of higher noise.

**X-ray Tube Current** reflects the X-ray intensity. Higher current typically produces cleaner, higher-contrast scans, whereas lower current leads to dimmer and noisier images.

These parameters are recorded in the RADCURE metadata and provide interpretable, reproducible axes for defining clinically realistic distribution shifts.

### 4.3 Data Partitioning Strategy

For each acquisition parameter, we first sort scans by the recorded value and group them according to distinct settings. We then construct an ID and OOD split according to the natural distribution of the target acquisition parameter, assigning the larger group to the ID domain to ensure sufficient training data and the smaller group to the OOD domain. For instance, for slice thickness, 2 mm scans (1,614) form the ID set and 2.5 mm scans (752) form the OOD set.

Fig. 4 visualizes the resulting separation for pixel spacing. Additional distributions for the other acquisition parameters are provided in Fig. 9 in the Appendix. The ID and OOD assignment rules are summarized

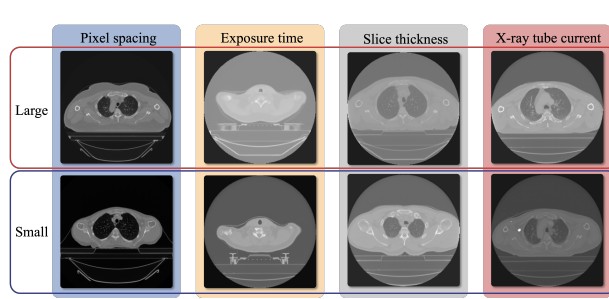

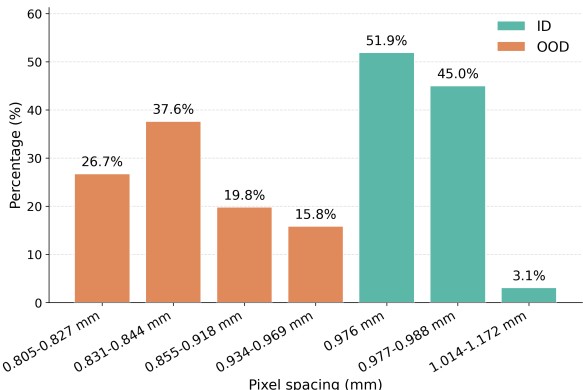

**Figure 3:** Examples showing how acquisition parameters affect CT image appearance. Larger pixel spacing covers a wider field of view with decreased resolution, longer exposure time and higher tube current reduce noise, and thicker slice thickness produces smoother but blurrier images.

**Figure 4:** Overall distribution of pixel spacing values across the RADCURE dataset. Percentages are normalized within each domain. The variability across parameter ranges enables a clear distinction between ID and OOD domains used in CURE-OOD.

in Table 2. For each task, samples without the corresponding acquisition metadata are excluded before partitioning. To ensure a unified evaluation protocol across acquisition-parameter splits, we use 1,462 ID training samples for each task, which is the maximum size consistently supported across all four splits.

### 4.4 A Survival-Aware Reference Baseline

Because MTLR does not produce standard classification-style confidence scores, classification-oriented OOD scores are not naturally aligned with its outputs. We therefore include HazRes as a simple survival-aware reference baseline defined on a survival-native quantity. The motivation is based on the assumption that the survival model is well fitted on the ID training distribution. When this holds, the model is expected to learn risk-related patterns from the ID training samples and produce hazard predictions that are stable and aligned with the empirical hazard profile of the training population. In survival prediction, lower-risk ID patients often share more typical risk-related patterns, while higher-risk patients may exhibit atypical features associated with worse outcomes. The model can therefore associate such atypical patterns with higher predicted hazard. When an input deviates from the ID distribution, it may activate similar risk-related patterns and lead to a higher predicted hazard. Motivated by this intuition, HazRes aggregates the signed difference between a test sample's predicted hazard curve and the average hazard curve on the training distribution, using this upward hazard shift as a survival-aware OOD signal.

**Training-phase statistics.** Given a trained MTLR model and the discrete-time hazard definition in Sec. 3.1, we first compute the hazard function $h_n(t_i)$ for each training sample $n$ at each discretized time bin $t_i$. We then calculate the average hazard over all $N$ training samples to obtain the expected hazard profile of the training distribution:

$$\bar{h}(t_i) = \frac{1}{N} \sum_{n=1}^{N} h_n(t_i). \tag{3}$$

**Testing-phase score.** For a test sample with hazard prediction $h_{\text{test}}(t_i)$, we measure its directional shift from the training distribution by computing the difference at each time bin and aggregating across all bins:

$$\text{HazRes}(\mathbf{x}) = \sum_{i=1}^{m} \left( h_\theta(t_i \mid \mathbf{x}) - \bar{h}(t_i) \right). \tag{4}$$

A larger HazRes value indicates that the sample's predicted hazard profile is shifted upward relative to the average hazard profile of the training population, serving as evidence of a potential distributional shift.

**Table 3:** OOD detection performance across survival tasks measured by AUROC and AUPRC. Average results summarize the relative performance of HazRes under acquisition shifts. 95% bootstrap confidence intervals are provided in Appendix Tables 8–9.

| Task | Method | Pixel Spacing | | Exposure Time | | Slice Thick. | | Tube Current | | Average | |
|---|---|---|---|---|---|---|---|---|---|---|---|
| | | AUROC | AUPRC | AUROC | AUPRC | AUROC | AUPRC | AUROC | AUPRC | AUROC | AUPRC |
| OS | ASH | 0.3898 | 0.4572 | 0.3683 | 0.4381 | 0.4513 | 0.4561 | 0.4504 | 0.5042 | 0.4150 | 0.4639 |
| | Dropout | 0.5000 | 0.4923 | 0.3660 | 0.4309 | 0.4396 | 0.4483 | 0.4685 | 0.5171 | 0.4435 | 0.4721 |
| | Energy | 0.3893 | 0.4550 | 0.3584 | 0.4310 | 0.4481 | 0.4914 | 0.4265 | 0.4394 | 0.4056 | 0.4542 |
| | GEN | 0.4972 | 0.4964 | 0.4057 | 0.4471 | 0.4326 | 0.4480 | 0.5522 | 0.5589 | 0.4719 | 0.4876 |
| | Dice | 0.4021 | 0.4639 | 0.3623 | 0.4301 | 0.4145 | 0.4431 | 0.4484 | 0.5101 | 0.4068 | 0.4618 |
| | MLS | 0.3927 | 0.4599 | 0.3612 | 0.4299 | 0.4485 | 0.4538 | 0.4461 | 0.5090 | 0.4121 | 0.4631 |
| | MSP | 0.5158 | 0.5085 | 0.4435 | 0.4624 | 0.4419 | 0.4516 | 0.5614 | 0.5654 | 0.4907 | 0.4970 |
| | ODIN | 0.5027 | 0.4983 | 0.3852 | 0.4356 | 0.4472 | 0.4535 | 0.5500 | 0.5536 | 0.4713 | 0.4853 |
| | SCALE | 0.3890 | 0.4560 | 0.3633 | 0.4369 | 0.4559 | 0.4580 | 0.4400 | 0.5058 | 0.4120 | 0.4642 |
| | NNGuide | 0.4353 | 0.4743 | 0.4295 | 0.4550 | 0.4874 | 0.4701 | 0.5012 | 0.5356 | 0.4634 | 0.4838 |
| | **HazRes** | **0.6124** | **0.5697** | **0.6401** | **0.6203** | **0.5519** | **0.5781** | **0.5701** | 0.5219 | **0.5936** | **0.5725** |
| LFFS | ASH | 0.3874 | 0.4377 | 0.3564 | 0.4101 | 0.4529 | 0.4694 | 0.4281 | 0.4853 | 0.4062 | 0.4506 |
| | Dropout | 0.4123 | 0.4443 | 0.3396 | 0.4009 | 0.4844 | 0.4875 | 0.4325 | 0.4951 | 0.4172 | 0.4570 |
| | Energy | 0.3822 | 0.4186 | 0.3428 | 0.4047 | 0.4921 | 0.5087 | 0.4489 | 0.4524 | 0.4165 | 0.4461 |
| | GEN | 0.5400 | 0.5046 | 0.4548 | 0.4484 | 0.4260 | 0.4563 | 0.4752 | 0.5042 | 0.4740 | 0.4784 |
| | Dice | 0.4117 | 0.4445 | 0.3439 | 0.4009 | 0.4775 | 0.4824 | 0.4319 | 0.4894 | 0.4163 | 0.4543 |
| | MLS | 0.4030 | 0.4422 | 0.3384 | 0.3994 | 0.4812 | 0.4837 | 0.4313 | 0.4889 | 0.4135 | 0.4535 |
| | MSP | 0.5011 | 0.4845 | 0.4427 | 0.4421 | 0.4299 | 0.4567 | 0.4556 | 0.4927 | 0.4573 | 0.4690 |
| | ODIN | 0.5066 | 0.4860 | 0.4112 | 0.4238 | 0.4742 | 0.4842 | 0.4523 | 0.4896 | 0.4611 | 0.4709 |
| | SCALE | 0.3888 | 0.4372 | 0.3398 | 0.4013 | 0.4879 | 0.4874 | 0.4288 | 0.4868 | 0.4113 | 0.4532 |
| | NNGuide | 0.4072 | 0.4438 | 0.3506 | 0.4035 | 0.4992 | 0.4909 | 0.5056 | **0.5264** | 0.4407 | 0.4662 |
| | **HazRes** | **0.6177** | **0.5991** | **0.6568** | **0.6694** | **0.5077** | **0.5370** | **0.5505** | 0.5239 | **0.5832** | **0.5824** |
| RFFS | ASH | 0.5053 | 0.5205 | 0.4116 | 0.4650 | 0.4529 | 0.4597 | 0.5329 | 0.5129 | 0.4757 | 0.4895 |
| | Dropout | 0.4856 | 0.4828 | 0.4033 | 0.4542 | 0.4509 | 0.4697 | 0.4260 | 0.4876 | 0.4415 | 0.4736 |
| | Energy | 0.4302 | 0.4644 | 0.4441 | 0.4959 | 0.4284 | 0.4662 | 0.4518 | 0.4566 | 0.4386 | 0.4708 |
| | GEN | 0.5707 | **0.5949** | 0.4954 | 0.4936 | 0.5843 | 0.5644 | **0.5695** | **0.5777** | 0.5550 | 0.5576 |
| | Dice | 0.4916 | 0.4831 | 0.3764 | 0.4454 | 0.4643 | 0.4750 | 0.4872 | 0.5086 | 0.4549 | 0.4780 |
| | MLS | 0.4770 | 0.4791 | 0.3970 | 0.4598 | 0.4603 | 0.4704 | 0.4287 | 0.4895 | 0.4408 | 0.4747 |
| | MSP | 0.5432 | 0.5650 | 0.5051 | 0.5125 | 0.6035 | 0.5833 | 0.4984 | 0.5197 | 0.5376 | 0.5451 |
| | ODIN | 0.5552 | 0.5833 | 0.4940 | 0.5168 | 0.5896 | 0.5838 | 0.4561 | 0.4875 | 0.5237 | 0.5429 |
| | SCALE | 0.5079 | 0.5266 | 0.4238 | 0.4631 | 0.4583 | 0.4659 | 0.5282 | 0.5234 | 0.4795 | 0.4947 |
| | NNGuide | 0.5624 | 0.5258 | 0.5435 | 0.5981 | 0.5716 | 0.5531 | 0.5635 | 0.5487 | 0.5603 | 0.5565 |
| | **HazRes** | **0.5720** | 0.5765 | **0.5560** | **0.6002** | 0.5716 | **0.5954** | 0.5480 | 0.5241 | **0.5619** | **0.5741** |
| DFFS | ASH | 0.4664 | 0.4799 | 0.4256 | 0.4346 | 0.4975 | 0.5230 | 0.5708 | **0.5600** | 0.4901 | 0.4994 |
| | Dropout | 0.5100 | 0.4973 | 0.4628 | 0.4541 | **0.5760** | **0.5645** | 0.4803 | 0.5020 | 0.5073 | 0.5045 |
| | Energy | 0.4689 | 0.5161 | 0.4304 | 0.4912 | 0.4635 | 0.4693 | 0.5294 | 0.5242 | 0.4730 | 0.5002 |
| | GEN | 0.4912 | 0.4868 | 0.4595 | 0.4652 | 0.5680 | 0.5552 | 0.5085 | 0.5354 | 0.5068 | 0.5106 |
| | Dice | 0.4658 | 0.4757 | 0.4361 | 0.4427 | 0.5007 | 0.5171 | 0.4666 | 0.5089 | 0.4673 | 0.4861 |
| | MLS | 0.4646 | 0.4785 | 0.4157 | 0.4323 | 0.4620 | 0.5051 | 0.5427 | 0.5381 | 0.4713 | 0.4885 |
| | MSP | 0.5007 | 0.4973 | 0.4501 | 0.4603 | 0.5610 | 0.5457 | 0.4887 | 0.5170 | 0.5001 | 0.5051 |
| | ODIN | 0.4923 | 0.4959 | 0.4305 | 0.4464 | 0.5153 | 0.5226 | 0.5111 | 0.5130 | 0.4873 | 0.4945 |
| | SCALE | 0.4644 | 0.4797 | 0.4287 | 0.4358 | 0.4826 | 0.5134 | 0.5632 | 0.5484 | 0.4847 | 0.4943 |
| | NNGuide | 0.5115 | 0.5075 | 0.4531 | 0.4486 | 0.5297 | 0.5390 | **0.5752** | 0.5557 | 0.5174 | 0.5127 |
| | **HazRes** | **0.5309** | **0.5362** | **0.5694** | **0.6058** | 0.5365 | 0.5159 | 0.4712 | 0.4907 | **0.5270** | **0.5371** |

## 5 Experiments

**Evaluation metrics.** We report metrics for both survival prediction and OOD detection. For survival prediction, we report Harrell's C-index. The risk score used for C-index evaluation is derived from the MTLR survival distribution by first computing the discrete hazard at each time interval and then summing the cumulative hazard over time. Endpoint- and split-specific censoring rates and censoring-handling details are reported in Appendix Table 6. For OOD detection, we use AUROC and AUPRC, measuring overall separability. Within each acquisition-shift setting, these metrics are computed by pooling the ID and OOD test cases.

**Survival prediction training.** To evaluate OOD detection under different distribution shifts, we train separate MTLR (Yu et al., 2011) survival models for each acquisition parameter–outcome pair. We consider four acquisition parameters and four outcomes (OS, LFFS, RFFS, DFFS), resulting in 16 models. Each model is trained on its corresponding ID training set and evaluated on both the matched ID test set and the paired OOD test set, reflecting deployment where acquisition conditions may change at inference time.

**Experimental setup.** We use a ViT-based 3D model with a UNETR (Hatamizadeh et al., 2022) encoder to extract volumetric features from CT, followed by two fully connected layers and an MTLR head that outputs survival distributions over discretized time. CT volumes are resampled to a common resolution and cropped around the primary tumor; the timeline is discretized into 8 intervals. We train with AdamW using a weight decay of 0.01 and a batch size of 32 for up to 400 epochs. Early stopping is applied based on validation loss with a patience of 20 epochs. We held out a fixed one fifth of the ID training cohort as

**Table 4:** Comparison of C-index between ID and OOD test sets across four survival prediction tasks. Dec. is computed as ID−OOD, where positive values indicate lower performance on OOD data. Most settings show reduced OOD performance, suggesting that acquisition-induced distribution shifts can affect survival prediction.

| Task | Pixel Spacing | | | Exposure Time | | | Slice Thickness | | | Tube Current | | |
|------|------|------|------|------|------|------|------|------|------|------|------|------|
| | ID | OOD | Dec. | ID | OOD | Dec. | ID | OOD | Dec. | ID | OOD | Dec. |
| OS | 0.7350 | 0.6428 | 0.0922 | 0.7977 | 0.6724 | 0.1253 | 0.7230 | 0.6642 | 0.0588 | 0.7185 | 0.6624 | 0.0561 |
| LFFS | 0.7166 | 0.5806 | 0.1360 | 0.6959 | 0.7137 | -0.0178 | 0.7823 | 0.6413 | 0.1410 | 0.8117 | 0.6840 | 0.1277 |
| RFFS | 0.7269 | 0.6487 | 0.0782 | 0.6309 | 0.6714 | -0.0405 | 0.7729 | 0.7612 | 0.0117 | 0.6986 | 0.4016 | 0.2970 |
| DFFS | 0.6555 | 0.5490 | 0.1065 | 0.7215 | 0.5621 | 0.1595 | 0.7113 | 0.4647 | 0.2466 | 0.6797 | 0.6804 | -0.0007 |

the validation set using a fixed random seed, and used the remaining four fifths for model training. The ID and OOD test sets were not used for training, tuning, early stopping, or selection. We use learning rates of $1{\times}10^{-3}$ for OS and LFFS, $8{\times}10^{-4}$ for RFFS, and $3{\times}10^{-4}$ for DFFS. Data augmentation includes random 3D rotations and shifts. All experiments are implemented in PyTorch and run on NVIDIA RTX A5000 GPUs.

# 6 Results on CURE-OOD

## 6.1 Performance Degradation under Data Shifts

To verify the impact of acquisition-induced shifts on downstream survival performance, we compare the C-index on the ID and OOD test sets for each acquisition parameter. Table 4 reports the C-index results across the four survival prediction tasks. Most acquisition-parameter splits exhibit lower C-index on OOD data than on ID data, with average decreases of 0.0831, 0.0967, 0.0866, and 0.1280 for OS, LFFS, RFFS, and DFFS, respectively. Several individual drops further exceed 0.1, indicating that the performance degradation can be substantial under certain acquisition shifts. Together, these degradation patterns suggest that distribution shifts can have measurable downstream consequences for survival prediction. The magnitude of degradation varies across both acquisition parameters and survival endpoints, suggesting that the effect of acquisition shift is not uniform but depends on both the imaging parameter being shifted and the clinical outcome being predicted.

## 6.2 Why Directly Adapted Classification-Style OOD Scores Can Fail

Classification-style OOD detectors are typically designed for class-confidence scores, whereas survival prediction produces structured, time-dependent outputs. In this setting, we treat the MTLR interval-wise logits $f_\Theta(\mathbf{x}, k)$ as class-logits and apply each conventional OOD scoring function according to its original definition to obtain reference baseline results. This provides a direct baseline comparison between naively adapted classification-style OOD scores and survival-aware alternatives on the same benchmark. In practice, Table 3 shows that several mainstream logit-based methods, including Energy, ASH, and SCALE (Xu et al., 2024), do not provide reliable OOD signals in survival prediction; they often show weak discrimination and, in some settings, even reversed ordering, where ID samples receive more OOD-like scores than OOD samples.

The reason is that survival logits do not encode class confidence in the same way as classification logits. Instead, as described in Sec. 3.1, they parameterize a time-ordered survival distribution and therefore reflect a structured, time-dependent risk profile. At early intervals, most patients have not yet experienced an event, so the training objective places greater emphasis on survival through those intervals, which in turn encourages the corresponding logits to move toward smaller, more negative values. Many classification-style OOD scores implicitly assume that larger logit magnitude or score values correspond to stronger in-distribution confidence. In survival prediction, however, larger logits are not necessarily more confident predictions, so the resulting score ordering need not follow the confidence assumptions used by classification-style OOD detectors.

Under acquisition-induced shift, OOD samples fall into less familiar feature regions, making the model more likely to assign higher event risk than it does for well-aligned ID samples. Their logits therefore become larger and drift toward zero. Consequently, ID samples often produce more negative logits than OOD samples, which is the opposite of the usual classification pattern where ID examples yield stronger, larger-magnitude

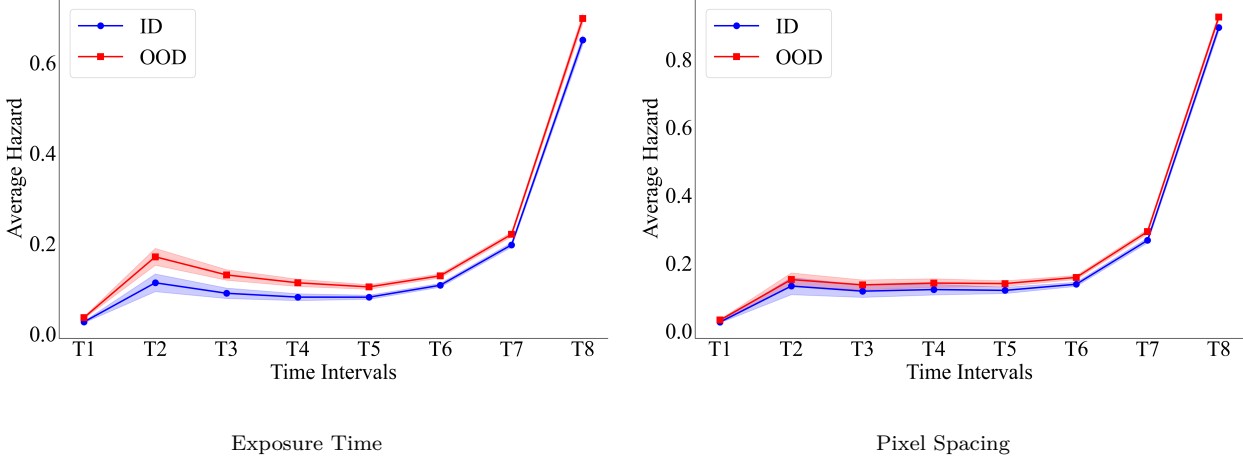

**Figure 5:** Mean hazard curves for ID and OOD test sets on OS under two acquisition shifts. Shaded regions indicate ±1 std. OOD curves exhibit higher mean hazard than ID curves in these examples.

evidence for the predicted class. When classification-style scores are applied on top of these logits, the ordering can invert: scores that are supposed to be larger for OOD inputs may instead become larger for ID inputs, causing systematic ID-as-OOD errors.

This effect is shown in Fig. 6, where the average OOD logits are consistently higher and closer to zero than the corresponding ID logits under a representative acquisition shift. This is fundamentally different from standard classification, where ID samples typically induce stronger class evidence and more extreme logits than OOD samples. The logits distributions in Fig. 7 and the gradient analysis in Sec. A further support the same mechanism. Together, these results explain why the evaluated classification-oriented OOD scores under this direct adaptation protocol are not reliably transferable to survival prediction, even when the same methods are effective in standard classification benchmarks.

### 6.3 Hazard-Based OOD Detection

In contrast, HazRes remains aligned with the output semantics of survival prediction by operating on the predicted hazard curve instead of raw logits. As shown in Table 3, it provides a competitive survival-aware reference among the compared methods across survival outcomes, with average AUROC improvements of 0.103, 0.109, 0.002, and 0.010 for OS, LFFS, RFFS, and DFFS, respectively, and average AUPRC improvements of 0.076, 0.104, 0.017, and 0.024 in the same order. These results suggest that hazard shift is a useful survival-aligned OOD signal under survival formulations.

To further probe this behavior, Fig. 5 visualizes the mean hazard curves on the ID and OOD test sets for OS under pixel spacing and exposure time shifts. OOD samples exhibit systematically higher mean hazard than ID samples, supporting the use of hazard shift as a task-consistent OOD signal. This suggests that future OOD methods for survival prediction should be defined on survival-native predictive quantities rather than directly transferred classification confidence scores.

To enable a broader comparison with survival-native baselines, we provide additional hazard-based methods in Appendix E. The results show that HazRes remains consistently competitive across these methods.

### 6.4 Architecture Sensitivity Analysis

To explore HazRes's performance with a different feature extractor, we use a ResNet model and repeat training and OOD detection on the OS task under the same four acquisition shifts. Table 5 shows that the overall ranking changes with the feature extractor, which is expected because different backbones induce different feature geometries for OOD scoring. However, the main qualitative trend remains consistent.

**Table 5:** OOD detection results on the OS task with a ResNet backbone. We report AUROC and AUPRC across four acquisition shifts.

| Method | Pixel Spacing | | Exposure Time | | Slice Thick. | | Tube Current | | Average | |
|---|---|---|---|---|---|---|---|---|---|---|
| | AUROC | AUPRC | AUROC | AUPRC | AUROC | AUPRC | AUROC | AUPRC | AUROC | AUPRC |
| ASH | 0.4452 | 0.4834 | 0.3680 | 0.4257 | 0.4198 | 0.4445 | 0.4765 | 0.4856 | 0.4274 | 0.4598 |
| Dropout | 0.4118 | 0.4513 | 0.3643 | 0.4247 | 0.4193 | 0.4434 | 0.4756 | 0.4853 | 0.4178 | 0.4512 |
| GEN | 0.4237 | 0.4655 | 0.3637 | 0.4242 | 0.4271 | 0.4486 | 0.4889 | 0.4997 | 0.4258 | 0.4595 |
| Energy | 0.4688 | 0.5332 | 0.3725 | 0.4271 | 0.4254 | 0.4791 | 0.4417 | 0.4477 | 0.4271 | 0.4717 |
| Dice | 0.4278 | 0.4709 | 0.3643 | 0.4241 | 0.4190 | 0.4437 | 0.4808 | 0.4891 | 0.4230 | 0.4569 |
| MLS | 0.4674 | 0.4861 | 0.3713 | 0.4260 | 0.4257 | 0.4461 | 0.4427 | 0.4743 | 0.4268 | 0.4581 |
| MSP | 0.4181 | 0.4636 | 0.3961 | 0.4368 | 0.4443 | 0.4563 | 0.4859 | 0.5005 | 0.4361 | 0.4643 |
| ODIN | 0.4244 | 0.4588 | 0.3836 | 0.4310 | 0.4338 | 0.4525 | 0.4865 | 0.5007 | 0.4321 | 0.4607 |
| SCALE | 0.4530 | 0.4817 | 0.3706 | 0.4260 | 0.4252 | 0.4458 | 0.4415 | 0.4751 | 0.4226 | 0.4572 |
| NNGuide | 0.4800 | 0.4936 | 0.3773 | 0.4281 | 0.4317 | 0.4497 | 0.4425 | 0.4745 | 0.4329 | 0.4615 |
| **HazRes** | **0.5312** | **0.5434** | **0.6276** | **0.6119** | **0.5746** | **0.5863** | **0.5581** | **0.5483** | **0.5729** | **0.5724** |

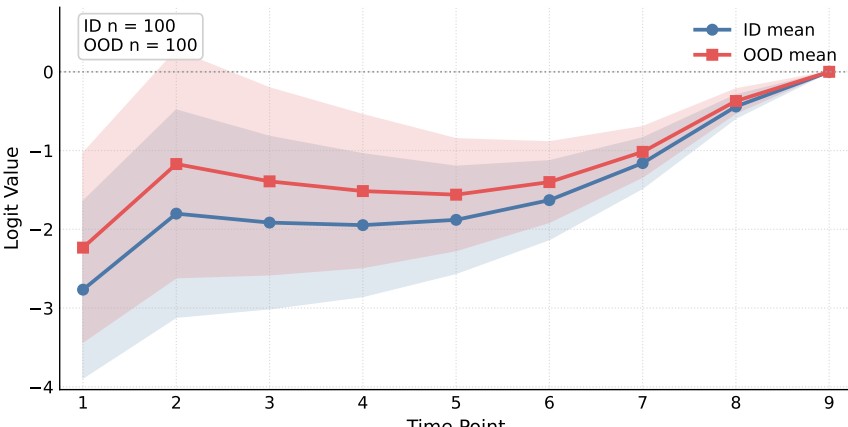

**Figure 6:** Mean logits on the OS task under the exposure time shift. Unlike standard classification logits, ID samples produce systematically smaller, more negative logits across time intervals, while OOD samples drift toward zero. This reversed ordering breaks the confidence interpretation assumed by classification-style OOD scores.

Hazard-based scores remain competitive after the backbone change and still provide a useful detection signal under acquisition shifts.

More specifically, the HazRes score remains particularly effective on exposure time and slice thickness. Among the methods retained from Table 3, it also attains the highest average AUROC and AUPRC. These results indicate that the utility of hazard shift is not tied to a single backbone design and that this survival-aware signal remains informative under a different feature extractor.

## 6.5   OOD Detectability and Downstream Survival Risk

Although most acquisition shifts lead to lower survival prediction performance on OOD test sets, OOD detection performance does not necessarily reflect how much an acquisition shift harms downstream survival prediction. Comparing Tables 3 and 4, we observe that OOD detection performance does not consistently track the C-index drop between ID and OOD test sets. For example, HazRes achieves high AUROC/AUPRC on LFFS under the exposure time shift, where the OOD C-index does not decrease. Conversely, RFFS under the tube current shift and DFFS under the slice thickness shift show large C-index drops, while their HazRes AUROC values remain only moderate.

These results indicate that acquisition-induced OOD shifts and downstream survival degradation are related but distinct aspects of deployment reliability. The features that make a sample distinguishable as OOD may only partially overlap with the features that make the survival prediction unreliable. Therefore, detecting a distribution shift is not equivalent to detecting a shift that harms survival prediction. A more useful OOD detector for survival prediction should preferentially capture shifts that perturb survival-relevant representations.

# 7    Conclusion and Discussion

We introduced CURE-OOD, to our knowledge, the first benchmark focused specifically on acquisition-induced OOD detection for cancer survival prediction from real-world clinical CT data, through a controlled evaluation protocol that uses acquisition-parameter variation to instantiate realistic covariate shifts. Through experiments spanning multiple acquisition shifts and representative OOD detectors, we find that acquisition shifts can degrade survival prediction, and that directly adapted classification-style OOD scores can be unreliable due to distinct logit dynamics. As a simple survival-aware reference baseline, HazRes further suggests that hazard-based signals can provide a useful direction for OOD detection in survival prediction. One limitation is that HazRes only captures upward hazard shifts and does not detect OOD samples with abnormally low predicted hazard. Future work may build on CURE-OOD to develop more task-aligned OOD detection and robustness evaluation under distribution shifts.

# Broader Impact Statement

CURE-OOD is intended as a research benchmark for studying OOD detection in cancer survival prediction, and is not a clinically validated tool. By making acquisition-induced distribution shifts explicit, the benchmark provides a controlled setting for evaluating whether reliability and OOD detection methods remain informative under clinically relevant changes in imaging acquisition.

**Clinical scope and risk of false confidence.** Survival predictions may influence risk stratification and treatment planning, so an unreliable model that is not flagged as operating outside its training distribution could contribute to harmful decisions. Our results show that mainstream classification-style OOD detectors can be unreliable for survival models, and that OOD detectability is only weakly correlated with downstream survival degradation. The absence of an OOD warning therefore does not establish that a prediction is accurate, calibrated, or clinically actionable. Any practical use would require prospective, regulated validation and careful integration into clinical workflows.

**Representativeness and fairness.** CURE-OOD is derived from RADCURE, one of the most extensive publicly accessible head and neck cancer imaging datasets, which includes patients treated between 2005 and 2017. Although it includes heterogeneity in scanner manufacturers and acquisition protocols, it may not reflect the full diversity of clinical sites, cancer types, modalities, and patient populations. Our experiments are also based on an MTLR survival model, so the findings may not necessarily generalize to other survival model architectures. Models developed on this benchmark should therefore be audited for subgroup and acquisition-condition disparities before any practical use.

**Communication of uncertainty.** OOD or uncertainty signals should not be treated as opaque scores. For such signals to be clinically useful, they must be calibrated, interpretable, and communicated to clinicians in an actionable way. How warnings should be surfaced and acted upon in clinical workflows is outside the scope of this benchmark and warrants dedicated human-factors and clinical study.

**Data governance and privacy.** RADCURE is publicly distributed through The Cancer Imaging Archive (TCIA), with access subject to data use terms, and the imaging data are de-identified by the data providers. We used the data in accordance with these terms for research purposes and add no information intended to re-identify patients.

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

## Appendix

## A    Gradient Dynamics of MTLR and Logit Behavior

To understand why logits-based OOD detection methods such as Energy and SCALE behave inversely on survival modeling tasks, we analyze the formulation of the MTLR model.

MTLR divides the continuous survival time into $m$ discrete intervals $\{t_1, t_2, \ldots, t_m\}$ and models the conditional probability of survival at each interval. For a patient with feature $\mathbf{x}$ and survival status sequence $y = (y_1, y_2, \ldots, y_m)$, where $y_i = 0$ indicates survival up to $t_i$, the likelihood is defined as:

$$P_\Theta(Y = y \mid \mathbf{x}) = \frac{\exp\big(\sum_{i=1}^m y_i(\theta_i^\top \mathbf{x} + b_i)\big)}{\sum_{k=0}^m \exp(f_\Theta(\mathbf{x}, k))}, \tag{5}$$

$$\text{where } f_\Theta(\mathbf{x}, k) = \sum_{i=k+1}^m (\theta_i^\top \mathbf{x} + b_i).$$

Here, $f_\Theta(\mathbf{x}, k)$ represents the unnormalized logit (score) associated with the event occurring at time interval $k$. The denominator normalizes these scores over all possible event times, forming a valid probability distribution across discrete survival intervals.

The model is optimized by minimizing the *regularized negative log-likelihood*, which corresponds to maximizing the log-likelihood of the observed survival sequences with additional $\ell_2$ and smoothness regularization terms (Yu et al., 2011). Formally, the optimization objective can be written as:

$$\min_\Theta \frac{C_1}{2} \sum_{j=1}^m \|\boldsymbol{\theta}_j\|_2^2 + \frac{C_2}{2} \sum_{j=1}^{m-1} \|\boldsymbol{\theta}_{j+1} - \boldsymbol{\theta}_j\|_2^2 - \mathcal{L}_{\text{MTLR}}, \tag{6}$$

where $\mathcal{L}_{\text{MTLR}}$ denotes the log-likelihood term:

$$\mathcal{L}_{\text{MTLR}} = \sum_{i=1}^N \Big[ \sum_{j=1}^m y_j(s_i)(\theta_j^\top \mathbf{x}_i + b_j)$$

$$- \log \sum_{k=0}^m \exp f_\Theta(\mathbf{x}_i, k) \Big]. \tag{7}$$

The first regularizer $\sum_j \|\boldsymbol{\theta}_j\|_2^2$ constrains the parameter magnitude to prevent overfitting, while the second term $\sum_j \|\boldsymbol{\theta}_{j+1} - \boldsymbol{\theta}_j\|_2^2$ enforces smoothness across consecutive time intervals.

During optimization, the first component of $\mathcal{L}_{\text{MTLR}}$ adjusts the scores associated with the observed survival sequence, whereas the second term normalizes the probabilities across all possible time sequences. This formulation calibrates the probability mass assigned to different event-time intervals. At early time intervals, where most samples remain alive, the model is encouraged to assign low probability mass to early event intervals. In our trained models, this probability-level calibration is empirically accompanied by smaller (more negative) logits at early intervals and larger (less negative) logits at later intervals. This trend is consistently observed across acquisition shifts in our experiments, as shown in Fig. 7, where the mean logits shift upward from early to late time bins. We analyze the gradient dynamics below to clarify this calibration mechanism and relate it to the observed empirical logit pattern.

### A.1    Gradient Analysis and Event-Time Probability Calibration

We derive the gradient of the core likelihood term and show how optimization calibrates the cumulative probability mass assigned to earlier event intervals. This derivation should be interpreted as a probability-level calibration argument rather than a proof that every SGD step decreases every early logit.

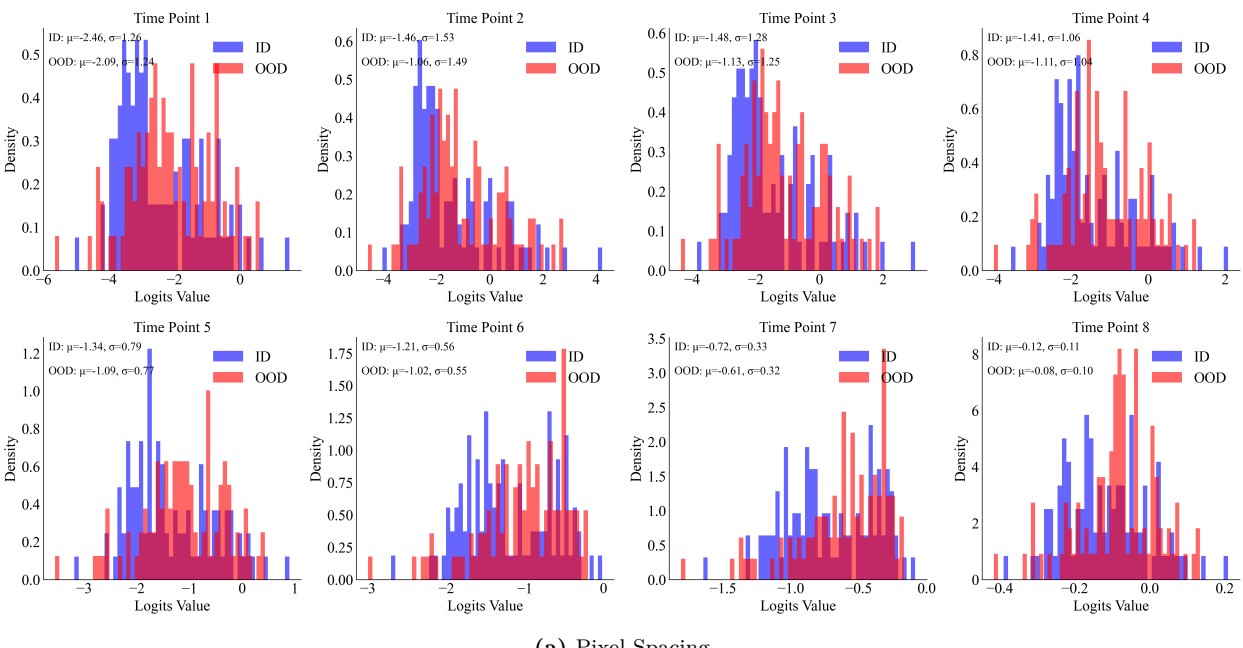

**(a)** Pixel Spacing

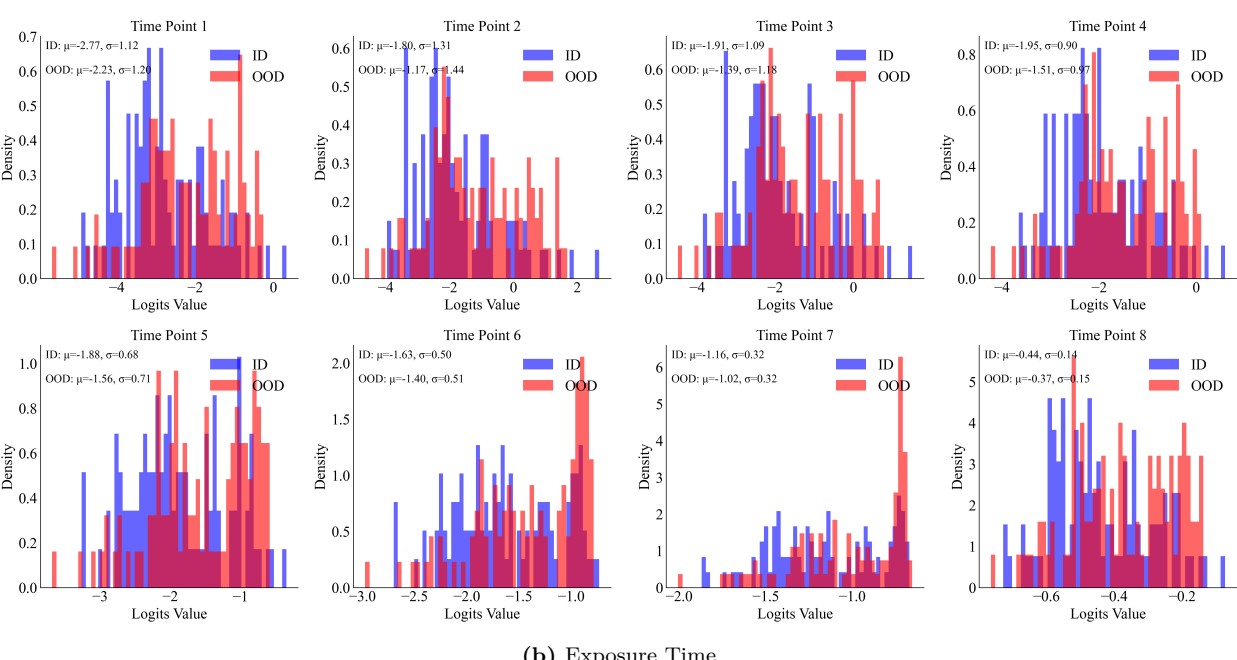

**(b)** Exposure Time

**Figure 7:** Visualization of MTLR logits distributions on the OS task under two acquisition shifts: (a) pixel spacing and (b) exposure time. For both shifts, ID samples tend to produce logits concentrated at lower values, whereas OOD samples show a clear tendency toward larger logits. This shift in the logits distribution indicates that acquisition differences systematically affect the model's output logit distribution, providing a useful signal for OOD detection.

**Setup.** For a sample $(\mathbf{x}_i, s_i)$, define the normalizer

$$S_i(\Theta) := \sum_{k=0}^{m} \exp f_\Theta(\mathbf{x}_i, k), \qquad \pi_{ik} := \frac{\exp f_\Theta(\mathbf{x}_i, k)}{S_i(\Theta)}.$$

Recall that $f_\Theta(\mathbf{x}, k) = \sum_{r=k+1}^m (\boldsymbol{\theta}_r^\top \mathbf{x} + b_r)$, so for a fixed $j \in \{1, \ldots, m\}$,

$$\frac{\partial f_\Theta(\mathbf{x}_i, k)}{\partial \boldsymbol{\theta}_j} = \begin{cases} \mathbf{x}_i, & k < j, \\ \mathbf{0}, & k \geq j. \end{cases} \tag{A.1}$$

**Gradient of the per-sample objective.** For the core likelihood term

$$\ell_i(\Theta) = \sum_{r=1}^m y_r(s_i)\, (\boldsymbol{\theta}_r^\top \mathbf{x}_i + b_r) - \log S_i(\Theta),$$

the gradient with respect to $\boldsymbol{\theta}_j$ is

$$\frac{\partial \ell_i(\Theta)}{\partial \boldsymbol{\theta}_j} = y_j(s_i)\, \mathbf{x}_i - \frac{\partial}{\partial \boldsymbol{\theta}_j} \log S_i(\Theta). \tag{A.2}$$

The first term contributes according to the encoded target $y_j(s_i)$. The second term involves the derivative of the log-normalizer, which we compute next.

**Derivative of the log-normalizer.** Using equation A.1, we have

$$\frac{\partial}{\partial \boldsymbol{\theta}_j} \log S_i(\Theta) = \frac{1}{S_i(\Theta)} \sum_{k=0}^m \exp f_\Theta(\mathbf{x}_i, k)\, \frac{\partial f_\Theta(\mathbf{x}_i, k)}{\partial \boldsymbol{\theta}_j}$$

$$= \frac{1}{S_i(\Theta)} \sum_{k=0}^{j-1} \exp f_\Theta(\mathbf{x}_i, k)\, \mathbf{x}_i = \sum_{k=0}^{j-1} \pi_{ik}\, \mathbf{x}_i. \tag{A.3}$$

Substituting equation A.3 into equation A.2 yields

$$\frac{\partial \ell_i(\Theta)}{\partial \boldsymbol{\theta}_j} = \left( y_j(s_i) - \sum_{k=0}^{j-1} \pi_{ik} \right) \mathbf{x}_i. \tag{A.4}$$

Summing over all samples gives the overall gradient

$$\frac{\partial \mathcal{L}_{\mathrm{MTLR}}}{\partial \boldsymbol{\theta}_j} = \sum_{i=1}^N \left( y_j(s_i) - \sum_{k=0}^{j-1} \pi_{ik} \right) \mathbf{x}_i, \tag{A.5}$$

to which one may add the $\ell_2$ and smoothness regularizers from (Yu et al., 2011) if included.

**Interpretation.** Let

$$M_{ij} := \sum_{k=0}^{j-1} \pi_{ik}$$

denote the model's cumulative probability assigned to the event occurring before $t_j$. Then the gradient equation A.5 is controlled by the residual

$$y_j(s_i) - M_{ij}.$$

Thus, the likelihood term adjusts the model so that the predicted early cumulative event probability $M_{ij}$ moves toward the encoded target $y_j(s_i)$.

Under the convention stated above, $y_j(s_i) = 0$ indicates survival up to $t_j$. For such event-free samples, the residual is

$$y_j(s_i) - M_{ij} = -M_{ij} \leq 0,$$

so this likelihood component suppresses excessive probability mass assigned to event times before $t_j$. For samples whose encoded target at $t_j$ is positive, the residual can instead be positive, in which case the update has the opposite sign. Therefore, the gradient should be interpreted as a calibration mechanism for the early cumulative event probability, not as a universal statement that every SGD step decreases every early logit.

Aggregated over the dataset, this calibration can produce the empirical pattern in which early-interval logits become smaller (more negative) and later-interval logits become larger (less negative), consistent with increasing event risk. We provide a detailed derivation of this mechanism in Section A.2, showing when the update suppresses or increases the cumulative probability of early intervals.

## A.2 From the Gradient to Early-Interval Probability Calibration

Building upon the gradient expression above, we further examine when an optimization step suppresses or increases the early cumulative probability mass $\sum_{k<j} \pi_{ik}$.

We show that maximizing the likelihood calibrates the cumulative early probability $\sum_{k<j} \pi_{ik}$ toward the encoded target $y_j(s_i)$. When the model assigns too much early-event probability mass relative to this target, the update suppresses that mass; when it assigns too little, the update can increase it.

**SGD update on $\boldsymbol{\theta}_j$.** Consider one SGD step on sample $(\mathbf{x}_i, s_i)$ with stepsize $\eta > 0$:

$$\Delta\boldsymbol{\theta}_j = \eta \frac{\partial \ell_i}{\partial \boldsymbol{\theta}_j} = \eta\Big(y_j(s_i) - \sum_{k=0}^{j-1} \pi_{ik}\Big)\mathbf{x}_i, \tag{A.5a}$$

where $\pi_{ik} = \exp f_\Theta(\mathbf{x}_i, k)/\sum_r \exp f_\Theta(\mathbf{x}_i, r)$.

By equation A.1, a small parameter update $\Delta\boldsymbol{\theta}_j$ induces a corresponding change in the logits $f_\Theta(\mathbf{x}_i, k)$ through the chain rule:

$$\Delta f_\Theta(\mathbf{x}_i, k) \approx \Big\langle \nabla_{\boldsymbol{\theta}_j} f_\Theta(\mathbf{x}_i, k), \Delta\boldsymbol{\theta}_j \Big\rangle = \Big(\frac{\partial f_\Theta(\mathbf{x}_i, k)}{\partial \boldsymbol{\theta}_j}\Big)^\top \Delta\boldsymbol{\theta}_j$$
$$= \mathbf{x}_i^\top \Delta\boldsymbol{\theta}_j, \qquad k < j, \tag{A.6a}$$

since $\partial f_\Theta(\mathbf{x}_i, k)/\partial \boldsymbol{\theta}_j = \mathbf{x}_i$ only holds for $k < j$. Substituting the update rule from equation A.5a gives

$$\Delta f_\Theta(\mathbf{x}_i, k) = \mathbf{x}_i^\top \Big[\eta\Big(y_j(s_i) - \sum_{r=0}^{j-1} \pi_{ir}\Big)\mathbf{x}_i\Big]$$

$$= \eta\Big(y_j(s_i) - \sum_{r=0}^{j-1} \pi_{ir}\Big)\|\mathbf{x}_i\|_2^2, \qquad k < j, \tag{A.6}$$

and leaves $f_\Theta(\mathbf{x}_i, k)$ unchanged for $k \geq j$ since $\partial f_\Theta/\partial \boldsymbol{\theta}_j = \mathbf{0}$ in that case.

**Sign of the update at early $j$.** Equation A.6 shows that the sign of the logit update is determined by the residual

$$y_j(s_i) - \sum_{r=0}^{j-1} \pi_{ir}.$$

Since $\eta > 0$ and $\|\mathbf{x}_i\|_2^2 \geq 0$, we have

$$\mathrm{sign}(\Delta f_\Theta(\mathbf{x}_i, k)) = \mathrm{sign}\Big(y_j(s_i) - \sum_{r=0}^{j-1} \pi_{ir}\Big), \qquad k < j. \tag{A.7}$$

Therefore, equation A.6 does not by itself imply that early logits always decrease. If the residual is negative, then the affected early logits $f_\Theta(\mathbf{x}_i, k)$ with $k < j$ decrease. If the residual is positive, those logits increase.

For example, under the convention stated above, an event-free sample at $t_j$ has $y_j(s_i) = 0$, so the residual is

$$-\sum_{r<j} \pi_{ir} \leq 0,$$

and this likelihood component decreases the affected early logits. In contrast, if a sample or encoding produces $y_j(s_i) \approx 1$ while $\sum_{r<j} \pi_{ir} \approx \epsilon$, then equation A.6 gives a positive update proportional to $1 - \epsilon$. Thus, the correct conclusion is conditional: the likelihood update calibrates the early cumulative probability mass, rather than guaranteeing that every SGD step decreases all early logits.

**Effect on the early cumulative probability.** The softmax probabilities are monotone in their logits. Writing $\boldsymbol{f}_i = (f_\Theta(\mathbf{x}_i, 0), \ldots, f_\Theta(\mathbf{x}_i, m))$ and $\boldsymbol{\pi}_i = \mathrm{softmax}(\boldsymbol{f}_i)$, the Jacobian is

$$\frac{\partial \pi_{iq}}{\partial f_\Theta(\mathbf{x}_i, \ell)} = \pi_{iq}\big(\mathbb{1}\{q = \ell\} - \pi_{i\ell}\big). \tag{A.8}$$

Let

$$M_{ij} := \sum_{q<j} \pi_{iq}.$$

Using equation A.8, we obtain

$$\frac{\partial M_{ij}}{\partial f_\Theta(\mathbf{x}_i, \ell)} = \begin{cases} \pi_{i\ell}(1 - M_{ij}), & \ell < j, \\ -\pi_{i\ell}M_{ij}, & \ell \geq j. \end{cases}$$

If the update affects only the early logits $\ell < j$, as in the $\boldsymbol{\theta}_j$-component of the update above, then

$$\Delta M_{ij} \approx \sum_{\ell<j} \pi_{i\ell}(1 - M_{ij})\Delta f_\Theta(\mathbf{x}_i, \ell). \tag{A.9}$$

Therefore, the cumulative early probability decreases only when the net perturbation to the affected early logits is negative. Substituting equation A.6 into equation A.9 gives

$$\Delta M_{ij} \approx \eta \left( y_j(s_i) - M_{ij} \right) \|\mathbf{x}_i\|_2^2 M_{ij}(1 - M_{ij}). \tag{A.10}$$

Hence, the early cumulative probability decreases when $y_j(s_i) - M_{ij} < 0$ and increases when $y_j(s_i) - M_{ij} > 0$. This confirms that the gradient performs a calibration update on early event-time probability mass, rather than an unconditional suppression step.

**Population effect.** Aggregating such SGD steps over the dataset, the residual $y_j(s_i) - M_{ij}$ encourages the predicted cumulative early-event probability $M_{ij}$ to match the empirical event pattern at each time bin. At early intervals, where survival is prevalent, many event-free samples contribute negative residuals of the form $-M_{ij}$, suppressing excessive probability mass assigned to early event times. Samples with positive encoded targets can contribute the opposite sign, so the effect is an aggregate calibration tendency rather than a per-step guarantee.

In our trained models, this probability-level calibration is empirically accompanied by smaller, more negative logits at early intervals and larger, less negative logits at later intervals, yielding the monotonic trend reported in the main text.

### A.3 Why ID Logits Are More Negative

For ID samples, the model has been optimized to align its predicted event-time distribution with the empirical survival pattern of the training data. At early time points, the survival probability is high and the event likelihood is low, so likelihood optimization encourages low probability mass for early event intervals. In our trained models, this probability-level calibration is empirically reflected by logits $f_\Theta(\mathbf{x}, k)$ that are more negative at early intervals and gradually become less negative at later intervals. Thus, we treat this logit polarity as an empirical property of the trained MTLR models, rather than as a per-step monotonic guarantee from the gradient dynamics.

### A.4 Why OOD Logits Become Larger

When encountering OOD samples, the feature representations $\mathbf{x}$ deviate from the training distribution. In these unseen regions, the learned parameters $\{\theta_i, b_i\}$ may produce less calibrated responses, and the model can generate logits that are closer to zero. Formally, since $f_\Theta(\mathbf{x}, k)$ is a linear projection of $\mathbf{x}$,

$$f_\Theta(\mathbf{x}, k) = \sum_{i=k+1}^{m} (\theta_i^\top \mathbf{x} + b_i),$$

a shift in $\mathbf{x}$ toward regions unobserved during training can reduce the projection magnitude $|\theta_i^\top \mathbf{x}|$, effectively making logits less negative. This reflects increased predictive uncertainty and weaker separation between survival intervals.

### A.5 Consequence for Logit-Based OOD Detection

Most classification-based OOD detection methods, such as Energy or SCALE, assume that OOD samples produce logits with smaller magnitudes (lower confidence) than ID samples. However, in our trained MTLR

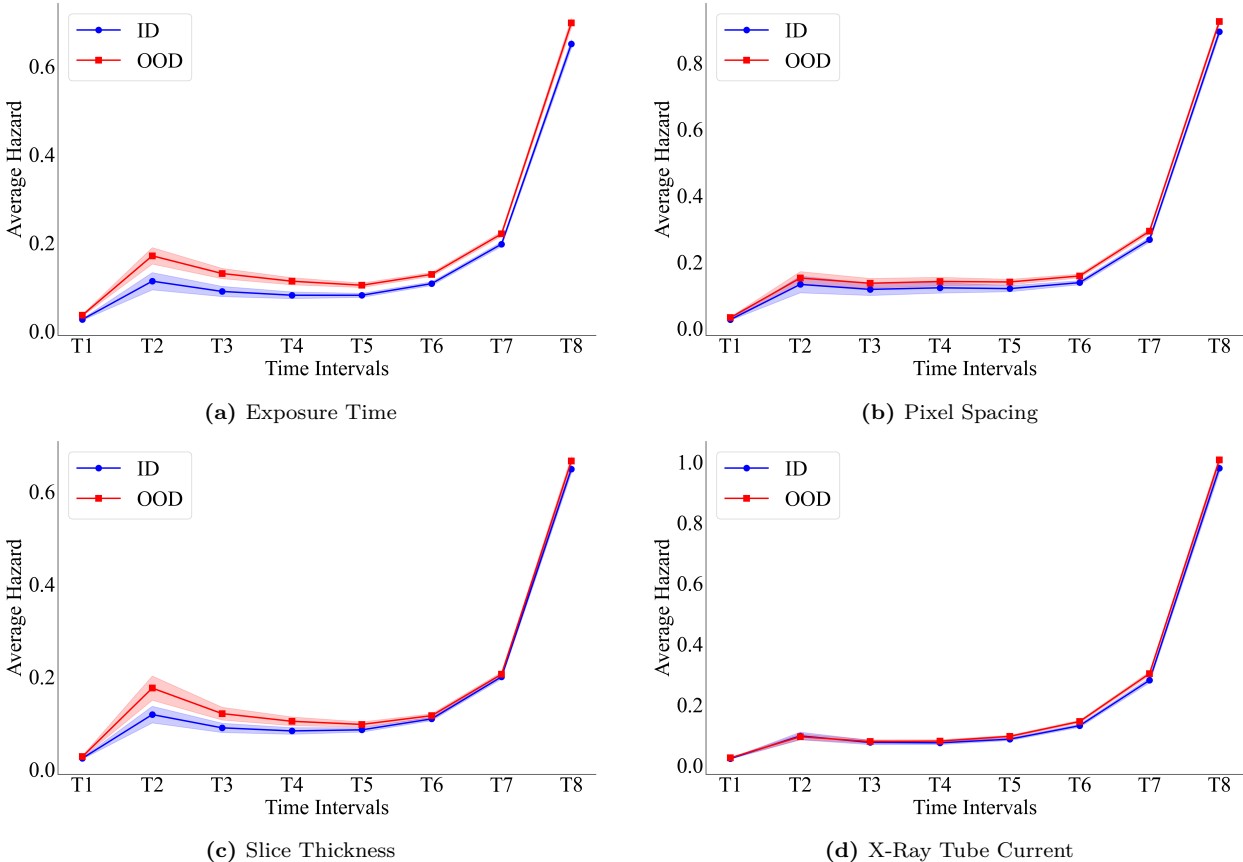

**Figure 8:** Mean hazard curves of ID and OOD test sets on the OS task across four acquisition shifts: (a) exposure time, (b) pixel spacing, (c) slice thickness, and (d) X-ray tube current. The shaded areas denote one standard deviation, showing the variability of predicted hazard values across samples. These plots show that acquisition-induced shifts can change the predicted hazard profiles, with OOD curves often shifted upward relative to ID curves.

survival models, we empirically observe a different polarity: ID samples tend to yield strongly negative logits due to their alignment with the high-survival regions of the training data, while OOD samples often result in less negative logits. Consequently, conventional logit-based energy scores can become inverted, causing ID samples to appear more OOD-like and OOD samples to appear more ID-like.

This inversion originates from the interplay between the monotonic survival constraint and distributional shifts in **x**. Empirically, we observe this consistent polarity reversal across most shifts in the CURE-OOD benchmark.

## B  Censoring Rates Across Outcomes and Acquisition Splits

Censoring is endpoint-specific because OS, LFFS, RFFS, and DFFS are defined using distinct event indicators. Consequently, within each acquisition split, censoring rates differ across outcomes. Table 6 reports the censoring rates for each outcome and acquisition split. Each ID and OOD test subset contains 100 patients. Event indicators are passed to the Harrell-style C-index computation as event-observed values. The reported C-index does not use IPCW weights.

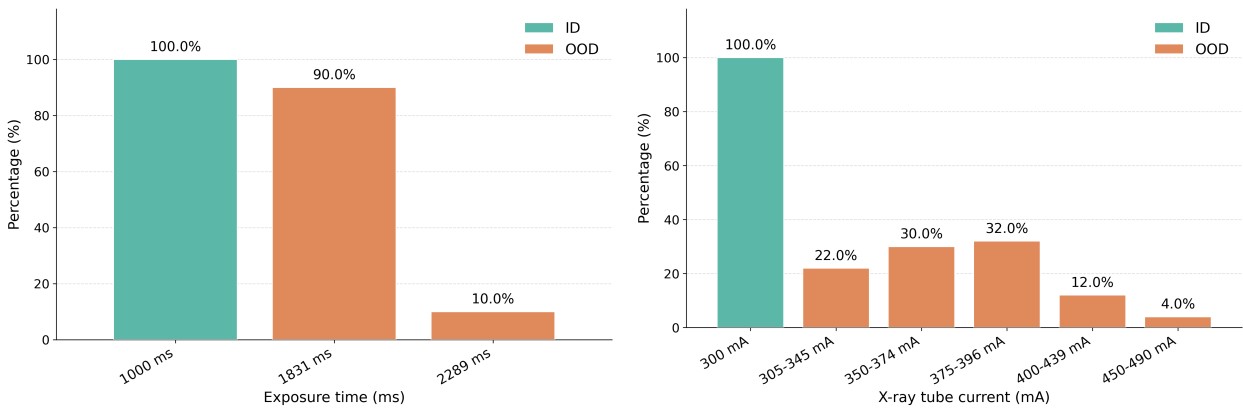

**(a)** Exposure time distribution across ID and OOD.  **(b)** X-ray tube current distribution across ID and OOD.

**Figure 9:** Distributions of key acquisition parameters used to construct ID and OOD domains in the CURE-OOD benchmark. Percentages are normalized within each domain. Each parameter exhibits distinct value ranges.

**Table 6:** Censoring rates for each survival outcome and acquisition split. Entries report the percentage of censored cases.

| Task | Pixel Spacing | | Exposure Time | | Slice Thickness | | Tube Current | |
|------|------|------|------|------|------|------|------|------|
| | ID | OOD | ID | OOD | ID | OOD | ID | OOD |
| OS | 63.0% | 49.0% | 79.0% | 52.0% | 73.0% | 51.0% | 75.0% | 67.0% |
| LFFS | 86.0% | 86.0% | 91.0% | 82.0% | 91.0% | 83.0% | 88.0% | 88.0% |
| RFFS | 95.0% | 93.0% | 96.0% | 90.0% | 90.0% | 93.0% | 96.0% | 97.0% |
| DFFS | 90.0% | 83.0% | 91.0% | 89.0% | 86.0% | 91.0% | 91.0% | 86.0% |

# C  Distribution-Aware Survival Prediction Evaluation

In addition to the C-index analysis, we further evaluate survival prediction performance using the integrated Brier score (IBS), which assesses the accuracy of the predicted survival probability distributions over time. To account for censoring, we compute IBS using an inverse-probability-of-censoring-weighted (IPCW) formulation, with the censoring distribution estimated nonparametrically within the corresponding evaluation split. This provides a distribution-aware complement to the main C-index results.

Table 7 compares IBS on the ID and OOD test sets across the four survival prediction tasks and four acquisition-parameter shifts. Overall, the OOD test sets show higher prediction error than the ID test sets, indicating that acquisition-induced distribution shifts degrade the quality of the predicted survival distributions. Averaged across acquisition shifts, the IBS increases by 0.0583 for OS, 0.1645 for LFFS, 0.1473 for RFFS, and 0.1236 for DFFS. The average trends are broadly consistent with the C-index degradation reported in the main text and provide additional evidence that OOD acquisition shifts negatively affect survival prediction performance from a distribution-aware perspective.

# D  Confidence intervals for OOD detection performance.

For each survival task, acquisition shift, and OOD detection method, we estimated uncertainty in AUROC and AUPRC using stratified nonparametric bootstrap resampling over ID and OOD test samples. The lower and upper bounds of the 95% confidence interval (CI) were obtained from the 2.5th and 97.5th percentiles of the resulting bootstrap distribution. As shown in Tables 8 and 9, HazRes maintains competitive performance across most survival tasks and acquisition shifts. Although some intervals overlap in more challenging settings, the overall pattern remains stable: HazRes generally attains higher point estimates than existing OOD baselines, supporting its robustness under acquisition-induced distribution shifts.

**Table 7:** Comparison of integrated Brier score (IBS) between ID and OOD test sets across survival prediction tasks. IBS is computed over the discretized MTLR time grid using IPCW-weighted Brier scores, with the censoring survival function estimated by Kaplan–Meier on the corresponding evaluation split. Lower values indicate more accurate survival distribution predictions. Δ is computed as OOD−ID, where positive values indicate higher prediction error on OOD data.

| Task | Pixel Spacing | | | Exposure Time | | | Slice Thickness | | | Tube Current | | |
|------|------|------|------|------|------|------|------|------|------|------|------|------|
| | ID | OOD | Δ | ID | OOD | Δ | ID | OOD | Δ | ID | OOD | Δ |
| OS | 0.2275 | 0.2280 | 0.0004 | 0.0954 | 0.2358 | 0.1404 | 0.0970 | 0.2420 | 0.1450 | 0.2778 | 0.2251 | -0.0527 |
| LFFS | 0.3742 | 0.4817 | 0.1075 | 0.0939 | 0.3712 | 0.2773 | 0.0952 | 0.3712 | 0.2760 | 0.3869 | 0.3840 | -0.0030 |
| RFFS | 0.4290 | 0.4132 | -0.0158 | 0.1007 | 0.4003 | 0.2996 | 0.0993 | 0.4030 | 0.3037 | 0.4501 | 0.4516 | 0.0015 |
| DFFS | 0.3615 | 0.3261 | -0.0354 | 0.0942 | 0.3526 | 0.2584 | 0.0975 | 0.3663 | 0.2688 | 0.3561 | 0.3588 | 0.0027 |

**Table 8:** AUROC for OOD detection across survival tasks and acquisition shifts. Entries report the point estimate with the corresponding 95% confidence interval.

| | Method | Pixel Spacing | Exposure Time | Slice Thick. | Tube Current |
|---|--------|---------------|---------------|--------------|--------------|
| OS | ASH | 0.3898 [0.3089, 0.4680] | 0.3683 [0.2969, 0.4464] | 0.4513 [0.3764, 0.5296] | 0.4504 [0.3682, 0.5275] |
| | Dropout | 0.5000 [0.4139, 0.5749] | 0.3660 [0.2944, 0.4448] | 0.4396 [0.3635, 0.5187] | 0.4685 [0.3874, 0.5421] |
| | Energy | 0.3893 [0.3100, 0.4701] | 0.3584 [0.2866, 0.4380] | 0.4481 [0.3714, 0.5249] | 0.4265 [0.3474, 0.5023] |
| | GEN | 0.4972 [0.4183, 0.5741] | 0.4057 [0.3307, 0.4840] | 0.4326 [0.3565, 0.5117] | 0.5522 [0.4711, 0.6293] |
| | Dice | 0.4021 [0.3212, 0.4798] | 0.3623 [0.2923, 0.4398] | 0.4145 [0.3340, 0.4922] | 0.4484 [0.3671, 0.5232] |
| | MLS | 0.3927 [0.3096, 0.4734] | 0.3612 [0.2902, 0.4391] | 0.4485 [0.3723, 0.5248] | 0.4461 [0.3642, 0.5228] |
| | MSP | 0.5158 [0.4366, 0.5981] | 0.4435 [0.3702, 0.5215] | 0.4419 [0.3648, 0.5198] | 0.5614 [0.4778, 0.6406] |
| | ODIN | 0.5027 [0.4243, 0.5785] | 0.3852 [0.3114, 0.4619] | 0.4472 [0.3665, 0.5247] | 0.5500 [0.4682, 0.6294] |
| | SCALE | 0.3890 [0.3080, 0.4690] | 0.3633 [0.2929, 0.4423] | 0.4559 [0.3823, 0.5322] | 0.4400 [0.3579, 0.5157] |
| | NNGuide | 0.4353 [0.3565, 0.5162] | 0.4295 [0.3522, 0.5109] | 0.4874 [0.4078, 0.5680] | 0.5012 [0.4213, 0.5776] |
| | **HazRes** | **0.6124** [0.5326, 0.6923] | **0.6401** [0.5596, 0.7118] | **0.5519** [0.4758, 0.6279] | **0.5701** [0.4926, 0.6511] |
| LFFS | ASH | 0.3874 [0.3057, 0.4602] | 0.3564 [0.2829, 0.4322] | 0.4529 [0.3779, 0.5348] | 0.4281 [0.3468, 0.5058] |
| | Dropout | 0.4123 [0.3303, 0.4911] | 0.3396 [0.2696, 0.4149] | 0.4844 [0.4144, 0.5625] | 0.4325 [0.3566, 0.5108] |
| | Energy | 0.3822 [0.3039, 0.4591] | 0.3428 [0.2740, 0.4207] | 0.4921 [0.4189, 0.5646] | 0.4489 [0.3679, 0.5235] |
| | GEN | 0.5400 [0.4573, 0.6206] | 0.4548 [0.3753, 0.5293] | 0.4260 [0.3521, 0.5043] | 0.4752 [0.3913, 0.5535] |
| | Dice | 0.4117 [0.3277, 0.4909] | 0.3439 [0.2718, 0.4187] | 0.4775 [0.4054, 0.5536] | 0.4319 [0.3537, 0.5114] |
| | MLS | 0.4030 [0.3215, 0.4811] | 0.3384 [0.2677, 0.4129] | 0.4812 [0.4102, 0.5552] | 0.4313 [0.3527, 0.5072] |
| | MSP | 0.5011 [0.4228, 0.5795] | 0.4427 [0.3648, 0.5187] | 0.4299 [0.3552, 0.5116] | 0.4556 [0.3780, 0.5330] |
| | ODIN | 0.5066 [0.4274, 0.5874] | 0.4112 [0.3336, 0.4862] | 0.4742 [0.3980, 0.5610] | 0.4523 [0.3695, 0.5303] |
| | SCALE | 0.3888 [0.3079, 0.4644] | 0.3398 [0.2705, 0.4119] | 0.4879 [0.4153, 0.5653] | 0.4288 [0.3489, 0.5059] |
| | NNGuide | 0.4072 [0.3266, 0.4842] | 0.3506 [0.2796, 0.4276] | 0.4992 [0.4273, 0.5740] | 0.5056 [0.4209, 0.5781] |
| | **HazRes** | **0.6177** [0.5408, 0.6961] | **0.6568** [0.5788, 0.7255] | **0.5077** [0.4351, 0.5810] | **0.5505** [0.4763, 0.6315] |
| RFFS | ASH | 0.5053 [0.4282, 0.5813] | 0.4116 [0.3328, 0.4897] | 0.4529 [0.3775, 0.5328] | 0.5329 [0.4545, 0.6161] |
| | Dropout | 0.4856 [0.4058, 0.5658] | 0.4033 [0.3241, 0.4790] | 0.4509 [0.3743, 0.5360] | 0.4260 [0.3463, 0.5034] |
| | Energy | 0.4302 [0.3557, 0.5101] | 0.4441 [0.3593, 0.5295] | 0.4284 [0.3491, 0.5071] | 0.4518 [0.3746, 0.5323] |
| | GEN | 0.5707 [0.4941, 0.6502] | 0.4954 [0.4128, 0.5726] | 0.5843 [0.5039, 0.6587] | **0.5695** [0.4860, 0.6455] |
| | Dice | 0.4916 [0.4128, 0.5679] | 0.3764 [0.2999, 0.4501] | 0.4643 [0.3861, 0.5426] | 0.4872 [0.4093, 0.5625] |
| | MLS | 0.4770 [0.4000, 0.5563] | 0.3970 [0.3172, 0.4755] | 0.4603 [0.3844, 0.5368] | 0.4287 [0.3500, 0.5044] |
| | MSP | 0.5432 [0.4618, 0.6258] | 0.5051 [0.4204, 0.5866] | **0.6035** [0.5227, 0.6815] | 0.4984 [0.4179, 0.5741] |
| | ODIN | 0.5552 [0.4756, 0.6336] | 0.4940 [0.4074, 0.5762] | 0.5896 [0.5068, 0.6649] | 0.4561 [0.3797, 0.5382] |
| | SCALE | 0.5079 [0.4290, 0.5871] | 0.4238 [0.3445, 0.5015] | 0.4583 [0.3764, 0.5426] | 0.5282 [0.4501, 0.6127] |
| | NNGuide | 0.5624 [0.4814, 0.6402] | 0.5435 [0.4646, 0.6188] | 0.5716 [0.4888, 0.6454] | 0.5635 [0.4867, 0.6405] |
| | **HazRes** | **0.5720** [0.4912, 0.6465] | **0.5560** [0.4713, 0.6402] | 0.5716 [0.4930, 0.6515] | 0.5480 [0.4683, 0.6258] |
| DFFS | ASH | 0.4664 [0.3839, 0.5481] | 0.4256 [0.3462, 0.5129] | 0.4975 [0.4209, 0.5754] | 0.5708 [0.4929, 0.6509] |
| | Dropout | 0.5100 [0.4353, 0.5935] | 0.4628 [0.3804, 0.5460] | **0.5760** [0.5013, 0.6573] | 0.4803 [0.4011, 0.5633] |
| | Energy | 0.4689 [0.3864, 0.5504] | 0.4304 [0.3557, 0.5162] | 0.4635 [0.3854, 0.5389] | 0.5294 [0.4569, 0.6147] |
| | GEN | 0.4912 [0.4132, 0.5741] | 0.4595 [0.3776, 0.5364] | 0.5680 [0.4901, 0.6436] | 0.5085 [0.4279, 0.5898] |
| | Dice | 0.4658 [0.3884, 0.5443] | 0.4361 [0.3565, 0.5149] | 0.5007 [0.4259, 0.5779] | 0.4666 [0.3896, 0.5491] |
| | MLS | 0.4646 [0.3865, 0.5469] | 0.4157 [0.3381, 0.5024] | 0.4620 [0.3851, 0.5368] | 0.5427 [0.4648, 0.6264] |
| | MSP | 0.5007 [0.4228, 0.5865] | 0.4501 [0.3686, 0.5264] | 0.5610 [0.4811, 0.6394] | 0.4887 [0.4087, 0.5713] |
| | ODIN | 0.4923 [0.4140, 0.5743] | 0.4305 [0.3552, 0.5064] | 0.5153 [0.4339, 0.5948] | 0.5111 [0.4339, 0.5949] |
| | SCALE | 0.4644 [0.3818, 0.5447] | 0.4287 [0.3525, 0.5161] | 0.4826 [0.4070, 0.5617] | 0.5632 [0.4863, 0.6444] |
| | NNGuide | 0.5115 [0.4301, 0.5919] | 0.4531 [0.3748, 0.5373] | 0.5297 [0.4516, 0.6079] | **0.5752** [0.5005, 0.6578] |
| | **HazRes** | **0.5309** [0.4491, 0.6135] | **0.5694** [0.4842, 0.6435] | 0.5365 [0.4604, 0.6143] | 0.4712 [0.3864, 0.5433] |

# E  Additional Survival-Native Baselines

To make the comparison more comprehensive, we include additional hazard-native OOD baselines in Table 10. These baselines are built directly from the predicted hazard curves and therefore serve as more natural survival-native references. Specifically, *HazardPos* uses the one-sided positive deviation from the average training hazard pattern, where positive deviations are retained and negative deviations are clipped to zero. *Hazard-L1* and *Hazard-L2* measure unsigned hazard-curve deviations using $\ell_1$ and $\ell_2$ distances

**Table 9:** AUPRC for OOD detection across survival tasks and acquisition shifts. Entries report the point estimate with the corresponding 95% confidence interval.

| | Method | Pixel Spacing | Exposure Time | Slice Thick. | Tube Current |
|---|---|---|---|---|---|
| OS | ASH | 0.4572 [0.4041, 0.5279] | 0.4381 [0.3958, 0.5036] | 0.4561 [0.4188, 0.5199] | 0.5042 [0.4493, 0.5840] |
| | Dropout | 0.4923 [0.4422, 0.5622] | 0.4309 [0.3905, 0.4929] | 0.4483 [0.4128, 0.5134] | 0.5171 [0.4524, 0.5869] |
| | Energy | 0.4550 [0.4061, 0.5266] | 0.4310 [0.3909, 0.4997] | 0.4914 [0.4367, 0.5695] | 0.4394 [0.4027, 0.5030] |
| | GEN | 0.4964 [0.4425, 0.5662] | 0.4471 [0.4036, 0.5113] | 0.4480 [0.4107, 0.5138] | 0.5589 [0.4905, 0.6344] |
| | Dice | 0.4639 [0.4088, 0.5348] | 0.4301 [0.3900, 0.4924] | 0.4431 [0.4050, 0.5070] | 0.5101 [0.4493, 0.5797] |
| | MLS | 0.4599 [0.4061, 0.5302] | 0.4299 [0.3899, 0.4918] | 0.4538 [0.4158, 0.5148] | 0.5090 [0.4467, 0.5802] |
| | MSP | 0.5085 [0.4520, 0.5823] | 0.4624 [0.4202, 0.5282] | 0.4516 [0.4145, 0.5174] | **0.5654** [0.4924, 0.6411] |
| | ODIN | 0.4983 [0.4463, 0.5689] | 0.4356 [0.3961, 0.4973] | 0.4535 [0.4169, 0.5179] | 0.5536 [0.4859, 0.6329] |
| | SCALE | 0.4560 [0.4033, 0.5281] | 0.4369 [0.3939, 0.5022] | 0.4580 [0.4186, 0.5222] | 0.5058 [0.4464, 0.5802] |
| | NNGuide | 0.4743 [0.4189, 0.5458] | 0.4550 [0.4119, 0.5172] | 0.4701 [0.4309, 0.5334] | 0.5356 [0.4739, 0.6078] |
| | **HazRes** | **0.5697** [0.5087, 0.6690] | **0.6203** [0.5497, 0.7162] | **0.5781** [0.5148, 0.6587] | 0.5219 [0.4745, 0.6029] |
| LFFS | ASH | 0.4377 [0.3963, 0.5045] | 0.4101 [0.3805, 0.4603] | 0.4694 [0.4255, 0.5441] | 0.4853 [0.4289, 0.5568] |
| | Dropout | 0.4443 [0.4030, 0.5116] | 0.4009 [0.3756, 0.4510] | 0.4875 [0.4425, 0.5646] | 0.4951 [0.4392, 0.5698] |
| | Energy | 0.4186 [0.3866, 0.4768] | 0.4047 [0.3774, 0.4601] | 0.5087 [0.4568, 0.5910] | 0.4524 [0.4131, 0.5195] |
| | GEN | 0.5046 [0.4565, 0.5805] | 0.4484 [0.4124, 0.5030] | 0.4563 [0.4154, 0.5335] | 0.5042 [0.4469, 0.5794] |
| | Dice | 0.4445 [0.4038, 0.5137] | 0.4009 [0.3761, 0.4508] | 0.4824 [0.4401, 0.5570] | 0.4894 [0.4357, 0.5618] |
| | MLS | 0.4422 [0.4005, 0.5111] | 0.3994 [0.3751, 0.4485] | 0.4837 [0.4414, 0.5599] | 0.4889 [0.4353, 0.5626] |
| | MSP | 0.4845 [0.4381, 0.5587] | 0.4421 [0.4078, 0.4968] | 0.4567 [0.4153, 0.5336] | 0.4927 [0.4390, 0.5665] |
| | ODIN | 0.4860 [0.4410, 0.5628] | 0.4238 [0.3943, 0.4702] | 0.4842 [0.4402, 0.5643] | 0.4896 [0.4359, 0.5634] |
| | SCALE | 0.4372 [0.3964, 0.5057] | 0.4013 [0.3753, 0.4521] | 0.4874 [0.4432, 0.5636] | 0.4868 [0.4321, 0.5600] |
| | NNGuide | 0.4438 [0.4023, 0.5143] | 0.4035 [0.3783, 0.4523] | 0.4909 [0.4459, 0.5655] | **0.5264** [0.4673, 0.6007] |
| | **HazRes** | **0.5991** [0.5327, 0.6814] | **0.6694** [0.5996, 0.7442] | **0.5370** [0.4776, 0.6114] | 0.5239 [0.4735, 0.6054] |
| RFFS | ASH | 0.5205 [0.4642, 0.6018] | 0.4650 [0.4180, 0.5385] | 0.4597 [0.4197, 0.5286] | 0.5129 [0.4583, 0.5851] |
| | Dropout | 0.4828 [0.4394, 0.5567] | 0.4542 [0.4087, 0.5277] | 0.4697 [0.4224, 0.5401] | 0.4876 [0.4331, 0.5704] |
| | Energy | 0.4644 [0.4180, 0.5363] | 0.4959 [0.4367, 0.5837] | 0.4662 [0.4186, 0.5387] | 0.4566 [0.4166, 0.5296] |
| | GEN | **0.5949** [0.5228, 0.6847] | 0.4936 [0.4402, 0.5778] | 0.5644 [0.5012, 0.6512] | **0.5777** [0.5114, 0.6727] |
| | Dice | 0.4831 [0.4398, 0.5565] | 0.4454 [0.4001, 0.5160] | 0.4750 [0.4275, 0.5458] | 0.5086 [0.4530, 0.5954] |
| | MLS | 0.4791 [0.4353, 0.5549] | 0.4598 [0.4123, 0.5381] | 0.4704 [0.4247, 0.5402] | 0.4895 [0.4327, 0.5720] |
| | MSP | 0.5650 [0.5001, 0.6509] | 0.5125 [0.4517, 0.5893] | 0.5833 [0.5191, 0.6742] | 0.5197 [0.4612, 0.6043] |
| | ODIN | 0.5833 [0.5138, 0.6772] | 0.5168 [0.4503, 0.6018] | 0.5838 [0.5171, 0.6655] | 0.4875 [0.4367, 0.5697] |
| | SCALE | 0.5266 [0.4702, 0.6130] | 0.4631 [0.4179, 0.5352] | 0.4659 [0.4213, 0.5354] | 0.5234 [0.4615, 0.6062] |
| | NNGuide | 0.5258 [0.4762, 0.6088] | 0.5981 [0.5348, 0.6872] | 0.5531 [0.4912, 0.6401] | 0.5487 [0.4898, 0.6373] |
| | **HazRes** | 0.5765 [0.5114, 0.6554] | **0.6002** [0.5245, 0.6868] | **0.5954** [0.5279, 0.6759] | 0.5241 [0.4714, 0.6065] |
| DFFS | ASH | 0.4799 [0.4341, 0.5618] | 0.4346 [0.4018, 0.4955] | 0.5230 [0.4623, 0.6025] | **0.5600** [0.4963, 0.6474] |
| | Dropout | 0.4973 [0.4517, 0.5840] | 0.4541 [0.4163, 0.5193] | **0.5645** [0.5002, 0.6435] | 0.5020 [0.4492, 0.5843] |
| | Energy | 0.5161 [0.4540, 0.5866] | 0.4912 [0.4350, 0.5704] | 0.4693 [0.4262, 0.5382] | 0.5242 [0.4719, 0.6147] |
| | GEN | 0.4868 [0.4418, 0.5675] | 0.4652 [0.4225, 0.5358] | 0.5552 [0.4970, 0.6392] | 0.5354 [0.4767, 0.6244] |
| | Dice | 0.4757 [0.4299, 0.5563] | 0.4427 [0.4061, 0.5051] | 0.5171 [0.4600, 0.5938] | 0.5089 [0.4539, 0.5937] |
| | MLS | 0.4785 [0.4332, 0.5583] | 0.4323 [0.3993, 0.4965] | 0.5051 [0.4450, 0.5810] | 0.5381 [0.4827, 0.6239] |
| | MSP | 0.4973 [0.4485, 0.5847] | 0.4603 [0.4179, 0.5286] | 0.5457 [0.4873, 0.6271] | 0.5170 [0.4570, 0.6009] |
| | ODIN | 0.4959 [0.4457, 0.5810] | 0.4464 [0.4096, 0.5108] | 0.5226 [0.4633, 0.5958] | 0.5130 [0.4607, 0.5947] |
| | SCALE | 0.4797 [0.4314, 0.5592] | 0.4358 [0.4032, 0.4981] | 0.5134 [0.4541, 0.5897] | 0.5484 [0.4893, 0.6367] |
| | NNGuide | 0.5075 [0.4565, 0.5956] | 0.4486 [0.4145, 0.5128] | 0.5390 [0.4759, 0.6197] | 0.5557 [0.4991, 0.6449] |
| | **HazRes** | **0.5362** [0.4765, 0.6190] | **0.6058** [0.5354, 0.6854] | 0.5159 [0.4669, 0.5994] | 0.4907 [0.4374, 0.5640] |

over discrete time intervals. These variants serve as survival-native baselines that directly use the temporal hazard predictions for OOD scoring.

Across tasks and acquisition shifts, HazRes achieves the best average AUROC and AUPRC in most settings, suggesting that incorporating structured hazard-shift information provides a stronger OOD signal than the alternative hazard-deviation baselines.

**Table 10:** Evaluation of survival-native hazard-deviation OOD scores. We compare HazardPos, unsigned $\ell_1/\ell_2$ hazard-curve deviation variants, and the proposed HazRes score. Average columns summarize performance across acquisition shifts. Bold and underlined entries denote the best and second-best variants, respectively.

| Method | Pixel Spacing | | Exposure Time | | Slice Thick. | | Tube Current | | Average | |
|---|---|---|---|---|---|---|---|---|---|---|
| | AUROC | AUPRC | AUROC | AUPRC | AUROC | AUPRC | AUROC | AUPRC | AUROC | AUPRC |
| **OS** HazardPos | 0.6040 | 0.5527 | 0.6311 | 0.6007 | 0.5516 | 0.5738 | 0.5317 | 0.5005 | 0.5796 | 0.5569 |
| Hazard-L1 | 0.4439 | 0.4947 | 0.5409 | 0.5707 | **0.5665** | **0.5841** | 0.3758 | 0.4214 | 0.4818 | 0.5177 |
| Hazard-L2 | 0.4462 | 0.4947 | 0.5440 | 0.5708 | 0.5628 | 0.5824 | 0.3739 | 0.4197 | 0.4817 | 0.5169 |
| HazRes | **0.6124** | **0.5697** | **0.6401** | **0.6203** | 0.5519 | 0.5781 | **0.5701** | **0.5219** | **0.5936** | **0.5725** |
| **LFFFS** HazardPos | 0.6057 | 0.5801 | 0.6501 | 0.6520 | 0.5038 | 0.5379 | 0.5145 | 0.5079 | 0.5685 | 0.5695 |
| Hazard-L1 | 0.4887 | 0.5331 | 0.5600 | 0.6200 | **0.5399** | **0.5530** | 0.4642 | 0.4704 | 0.5132 | 0.5441 |
| Hazard-L2 | 0.4900 | 0.5310 | 0.5640 | 0.6216 | 0.5304 | 0.5485 | 0.4655 | 0.4711 | 0.5125 | 0.5430 |
| HazRes | **0.6177** | **0.5991** | **0.6568** | **0.6694** | 0.5077 | 0.5370 | **0.5505** | **0.5239** | **0.5832** | **0.5824** |
| **RFFFS** HazardPos | 0.5534 | 0.5522 | 0.5570 | 0.6005 | 0.5554 | 0.5672 | 0.5103 | 0.5048 | 0.5440 | 0.5562 |
| Hazard-L1 | 0.5037 | 0.5460 | **0.6151** | 0.6217 | 0.5501 | 0.5853 | 0.4613 | 0.4831 | 0.5325 | 0.5590 |
| Hazard-L2 | 0.4986 | 0.5397 | 0.6114 | **0.6254** | 0.5508 | 0.5944 | 0.4594 | 0.4821 | 0.5300 | 0.5604 |
| HazRes | **0.5720** | **0.5765** | 0.5560 | 0.6002 | **0.5716** | **0.5954** | **0.5480** | **0.5241** | **0.5619** | **0.5741** |
| **DFFFS** HazardPos | 0.5293 | 0.5288 | 0.5663 | 0.5923 | 0.5129 | 0.4959 | 0.4979 | 0.5018 | 0.5266 | 0.5297 |
| Hazard-L1 | 0.5231 | 0.5338 | 0.5541 | 0.5970 | 0.4295 | 0.4683 | **0.5397** | **0.5234** | 0.5116 | 0.5306 |
| Hazard-L2 | 0.5196 | 0.5332 | 0.5537 | 0.5991 | 0.4185 | 0.4583 | 0.5357 | 0.5207 | 0.5069 | 0.5278 |
| HazRes | **0.5309** | **0.5362** | **0.5694** | **0.6058** | **0.5365** | **0.5159** | 0.4712 | 0.4907 | **0.5270** | **0.5371** |

