# OpenReview forum: "CURE-OOD: Benchmarking Out-of-Distribution Detection for Survival Prediction"
_TMLR — Under review for TMLR_

### Review · Reviewer_AKQv · 2026-05-18

**Summary Of Contributions:**

This paper introduces CURE-OOD, which is presented as the first benchmark for studying out-of-distribution detection in cancer survival prediction under CT imaging acquisition shifts. The work is practically meaningful because it focuses on OOD samples caused by differences in imaging acquisition settings, which are common in real clinical data for cancer survival prediction. The authors point out an important gap that although OOD detection has been widely studied for classification and segmentation tasks, it has received much less attention in survival prediction, where models must handle time-to-event outcomes and censored data. The benchmark is built from the publicly available RADCURE dataset and uses four CT acquisition parameters to create controlled and clinically realistic distribution shifts across training, in-distribution test, and OOD test sets. However, the benchmark is based on only one dataset and one survival model framework, so it is still unclear how well the findings generalize to other cancer types, or imaging modalities.

The authors also evaluate several commonly used OOD detection methods originally designed for classification tasks, including MSP, ODIN, Energy, ASH, and SCALE, in survival prediction settings. Their experiments show that these methods often do not work well with survival models based on MTLR outputs. The paper argues that survival model outputs have different temporal and probabilistic meanings compared to classification logits, which can lead to unstable or even reversed OOD scoring behavior. At the same time, OOD detection performance across methods is generally low, with many AUROC values close to random guessing, which makes it difficult to clearly separate strong and weak approaches.

In addition, the paper proposes HazardDev, a simple survival-specific baseline that measures how much predicted hazard curves differ from the average hazard pattern in the training data. Results show that HazardDev generally performs better than several standard post-hoc OOD detection methods across different tasks and acquisition shifts, suggesting that survival-specific predictive signals may be more suitable for OOD detection than traditional classification confidence scores. However, the improvement is still modest, and the benchmark test sets are relatively small, which may limit the reliability of some performance comparisons.

**Audience:**

Yes

**Audience Explanation:**

This paper would likely interest part of the TMLR audience, especially researchers working in medical machine learning and survival analysis. It studies an underexplored problem of OOD detection for survival prediction from medical images and is practically important as medical imaging data often change across scanners and acquisition settings, which can affect model reliability in real clinical use.

The paper may be less interesting to readers focused on more general OOD methods or stronger theoretical contributions, since the work is fairly domain-specific and the proposed HazardDev method is relatively simple with fairly weak OOD detection results across all methods, so the practical value may not be fully clear.

Overall, even though the paper may not appeal to the full TMLR audience, it addresses a meaningful and overlooked problem that should be valuable to researchers interested in medical AI and reliable machine learning.

**Broader Impact Concerns:**

The paper lacks a clear ethical implications of work statement, and given its clinical focus, several ethical issues should be addressed:
1. There is a risk of false confidence in clinical use. The paper shows that current OOD detectors perform poorly. If used in practice, they may fail to flag important distribution shifts, which could lead to unreliable survival predictions affecting treatment decisions.
2. The benchmark is based on a single dataset (RADCURE, head-and-neck cancer from one institution), which may not reflect the diversity of real-world patients, scanners, or clinical settings. This raises concerns about generalizability and possible unequal performance across different populations.
3. Survival predictions can strongly influence clinical decisions and patient psychology, but the paper does not discuss how uncertainty or OOD warnings should be communicated in practice.
4. While the dataset is publicly available, the authors should still clarify data governance and privacy safeguards.
The paper would benefit from a dedicated ethical implication statement discussing these risks and clearly stating that the benchmark is for research purposes, not clinical deployment.

**Claims And Evidence:**

Yes

**Claims Explanation:**

I think most of the paper’s claims are supported by clear and convincing experiments. The benchmark is carefully designed using real CT acquisition metadata, and the results consistently show that acquisition-related distribution shifts can reduce survival prediction performance. The paper also provides strong evidence that many classification-based OOD detection methods do not work well for survival prediction, supported by both experiments and theoretical explanations.

The proposed HazardDev baseline generally performs better than the compared methods, but the evidence for its usefulness is still limited. Its AUROC scores are relatively low, suggesting only modest ability to separate ID and OOD samples. In addition, the benchmark uses only one dataset and one survival model, and the test sets are small with no confidence intervals or significance tests reported. So the results are convincing, but the claim that HazardDev is broadly useful is only partly proven.

**Requested Changes:**

Authors should consider the following adjustments:
1. The authors should include stronger statistical testing, which is critical to support the reliability of the results for acceptance. Currently, the ID and OOD test sets are small, and the differences between methods are often minor. Because of this, it is unclear whether the AUROC and AUPRC differences are real or just due to random variation. The authors should at least report bootstrapped confidence intervals or run significance tests, or add a baseline with uncertainty-based OOD methods. This is critical because many conclusions depend on small performance differences used to rank methods.
2. It’s also critical to include more discussion on the weak link between OOD scores and clinical performance. The results show a very weak relationship between OOD detection results and C-index drops, but this is not discussed enough. If OOD detectors cannot identify shifts that actually harm survival performance, it questions the practical value of the benchmark. The authors should explain what this means, and what a good OOD detector for survival prediction should capture.
3. It’s recommended to improve generalization and evaluation scope of this study to strengthen the work. The study is limited to the MTLR model and uses only 100 samples per split. The authors could strengthen the work by testing at least one other survival model or better justifying the sample size and experimental setup. It would also help to expand discussion on whether results generalize beyond RADCURE and head-and-neck CT data, and beyond MTLR-based survival prediction.

---

> ### Author Response · Authors · 2026-06-28
>
> Thank you for the careful reading and the useful suggestions.
>
> Manuscript revisions: We added bootstrap confidence intervals for OOD detection performance and expanded the discussion of the weak relationship between OOD separability and downstream C-index drop. We also added a dedicated broader impact statement discussing false confidence in clinical use, representativeness and fairness, uncertainty communication, and data governance/privacy.
>
> 1. Stronger statistical testing. Thank you for the reminder. We now report 95% bootstrap confidence intervals for AUROC and AUPRC across all methods, tasks, and shifts in Appendix D, obtained using stratified nonparametric bootstrap. We also provided an uncertainty-based baseline (Dropout). The intervals show that our method has a clear advantage in several settings, even if some intervals overlap in the harder cases.
> 2. OOD scores and clinical performance. We have revised Sec. 6.5 to discuss this issue more explicitly. Our results show that acquisition-induced OOD shifts often degrade downstream survival prediction, but the performance of OOD detection methods do not show a strong connection with the C-index drop. Our benchmark design follows the standard OOD definition and evaluation protocol: the ID and OOD sets differ by controlled data shift, and the test set is not used for training or model selection. The baseline OOD detectors may fail because they were not designed for survival prediction. Our method didn’t show a strong connection because it is based on predicted hazards and may not capture the feature changes that directly harm downstream survival performance. A better OOD detector for survival prediction may need to use sample-level features and capture shifts that perturb survival-relevant representations. Our benchmark provides a foundation for developing more task-aligned OOD detection methods for survival prediction in future work.
>
> 3. Each task uses 100 ID and 100 OOD cases, which represent a non-trivial fraction of the available dataset. Given the limited size of the full cohort, using a larger test set would further reduce the data available for training. RADCURE is one of the largest publicly available head-and-neck cancer imaging cohorts and includes scans acquired with different scanners and acquisition protocols. Therefore, the benchmark reflects real acquisition diversity to some degree. We also agree that this evaluation is not intended to show that our findings necessarily hold for every cancer type or survival model. Our goal is to provide a controlled benchmark for studying acquisition-induced OOD detection in survival prediction. We also agree that it would be valuable to extend this analysis beyond RADCURE and head-and-neck CT, as well as beyond the MTLR survival head. Since our splitting and evaluation protocol does not depend on a specific dataset or model, it can be directly reused to build similar benchmarks for other cohorts and modalities, and to test other survival models. We view this as an important direction for future work, especially because benchmarks of this kind are still limited. These discussions are included in Broader Impact Statement. We have added a concise discussion of the generalization scope to the Broader Impact Statement.
> 4. Broader Impact. Thank you for the suggestion. We have added a Broader Impact Statement accordingly.

---

### Review · Reviewer_hu4u · 2026-06-10

**Summary Of Contributions:**

Variations in imaging acquisition in the medical field can introduce OOD samples caused by covariate shifts that undermine model reliability, and so far, this has not been systematically studied in survival analysis. The paper thus propose a benchmarking tool to evaluate out-of-distribution (OOD) error in survival analysis they call CURE-OOD. The benchmark is built from the RADCURE cohort and defines ID/OOD splits using scanner/acquisition parameters such as pixel spacing, exposure time, slice thickness, and tube current. The paper evaluates whether these shifts degrade downstream survival prediction and whether standard classification-style OOD detectors transfer to survival models. It also proposes HazardDev, a simple survival-aware OOD score based on deviations in predicted hazard curves. The problem is timely and relevant, especially for medical imaging, robustness, OOD detection, and survival analysis.

Strengths:
* Important and clinically relevant problem setting
* Useful benchmark direction for survival prediction under acquisition shifts.
* Good explanation of why survival prediction differs from standard classification.
* HazardDev is a simple and intuitive survival-aware reference baseline.

Weaknesses:
* Related work misses important survival-specific literature on distribution shift.
* Benchmark split construction is not sufficiently clear or reproducible.
* HazardDev is mathematically under-specified and, as written, does not properly measure curve deviation.
* Survival evaluation relies only on an unspecified C-index without censoring details.
* Hyperparameter tuning, validation splits, and model-selection procedures are unclear.
* OOD baseline adaptation from MTLR logits is under-specified, making the comparison hard to interpret.
* Results lack uncertainty estimates despite small ID/OOD test sets.

**Audience:**

Yes

**Audience Explanation:**

Yes. The topic is likely to interest a subset of TMLR’s audience, especially researchers working on robustness, OOD detection, medical imaging, and survival analysis. The paper studies a practically important problem: whether acquisition-induced CT shifts degrade survival prediction and whether existing OOD detectors are suitable for this setting. Even though I have concerns about the clarity and rigor of parts of the evaluation, the problem formulation and benchmark direction are relevant and potentially useful.

**Broader Impact Concerns:**

The work is motivated by clinically relevant survival prediction from CT imaging, so I think a Broader Impact Statement would be appropriate. My main concern is that OOD detection results could be overinterpreted as sufficient for safe deployment. In clinical settings, identifying acquisition-induced shift is useful, but an OOD score does not by itself establish that a survival prediction is reliable, calibrated, or clinically actionable. This is especially important because the paper itself finds that OOD detectability and downstream survival degradation are only weakly related.

**Claims And Evidence:**

No

**Claims Explanation:**

Partially. I don't think the current evidence is fully clear or convincing for several of the paper’s stronger claims. In particular, the benchmark split construction is not sufficiently reproducible, the proposed HazardDev score does not appear to measure deviation as written, the survival evaluation relies on an unspecified C-index without censoring details, and the adaptation of classification-style OOD baselines to MTLR logits is under-specified. The results also lack uncertainty estimates despite small ID/OOD test sets. Finally, the novelty framing is weakened by missing related work on survival analysis under distribution shift.

**Requested Changes:**

Critical:

* The related work section mainly talks about OOD outside of survival analysis and in other kinds of medical datasets, but I feel there should be a separate section discussed what has already been done in survival and how the present work relates to that. OOD has already been studied in this setting and, but right now the paper never touches on this. It leaves the reader feeling this is an entire new problem of the field, which it is not, and this weakens the novelty claiming that CURE-OOD is the first benchmark for systematically evaluating OOD detection in survival prediction. See, for example, papers [1]-[5].
* Table 2 needs a better caption. Where does the "ID" and "OOD" come from and are those the test cases?
Section 4.3 states that 100 cases are randomly sampled from each group to form the ID and OOD test sets, which appears to explain the fixed "100/100" entries in Table 2, but it is less clear how the reported training-set sizes are obtained. For example, the text states that for slice thickness, 2 mm scans form the ID set with 1,614 cases and 2.5 mm scans form the OOD set with 752 cases, but how did you end up with only 1,200 training instances and what happened to the missing patients?
* The proposed HazardDev baseline in Eq. 4 is useful, but I don't see any measure of deviation from the training hazard curve. Since the hazards are simply summed over m intervals, positive and negative deviations across time can cancel. As a result, a sample with a substantially different hazard shape could receive a small HazardDev score. Did you intend for HazardDev to be an L1/L2 standardized distance between the two curves? If the current signed sum is intentional, I believe the claim that larger values indicate greater OOD-ness is not valid. Also for Eq. 4, please make it explicit that the test hazard curve is model-predicted from x. Right now "x" is passed to HazardDev, but not used. If you want to keep the signed sum, I would write something like HazardDev(x) = \sum_{i=1}^m (h_\theta(t_i | x) - h̄(t_i)). For actual deviation, you want something like HazardDev_L1(x) = sum_{i=1}^m |h_theta(t_i | x) - h_bar(t_i)|
* Section 5: You refer to "survival prediction" performance but evaluate it only using the C-index. Since MTLR outputs discrete survival distributions, ranking performance alone does not assess whether the predicted survival probabilities or event-time distributions are accurate (see, for example, recent work by [6]). MTLR gives a nice survival distribution, but you are only evaluating a small portion of that -- a derived risk score at a certain time point -- and it only evaluate ranking performance, not anything else. Either give a clear motivation here for why the C-index makes the most sense for OOD and how you derive it, or consider additional distribution-aware metrics such as the integrated Brier score, time-dependent Brier score, calibration, or survival-time error (MAE).
* Section 5: Saying "we use the concordance index" is incomplete. There are multiple variants -- Harrell's C-index, Uno's IPCW C-index, Antolini's C-index, Time-dependent AUC. This matters because right-censoring can affect concordance estimates, and Harrell’s C-index is known to be sensitive to censoring. Please specify the exact C-index implementation, how risk scores are derived from the MTLR survival distribution, and whether any adjustment for censoring is applied (eg, IPCW). It would also be useful to report censoring rates for each outcome and split, especially since ID and OOD domains may differ in censoring distributions.
* Section 5: The hyperparameter procedure is not described in detail. You report optimizer settings, learning rates, early stopping, and augmentation, but it is unclear whether these choices were tuned or fixed to begin with. Can you clarify please what validation set was used for early stopping and hyperparameter selection, whether validation data came only from the ID training distribution, and whether OOD test performance was ever used during model selection.
* Section 5, Tab 3: I believe this reports OOD detection performance by pooling the 100 ID and 100 OOD test samples and computing AUROC/AUPRC from the resulting binary ID-vs-OOD labels. If yes, write that explicitly, since you do not compute them separately for ID and OOD domains. It would also be very good to report confidence intervals or standard deviations here, because each OOD detection evaluation only use 200 samples (which is already a very small number). In Section 6.3, you report average AUROC improvements as low as 0.004 for DFFS, which is not really meaningful without any uncertainty.
* Section 6.2: This comparison is under-specified. How did you go from interval-wise logits to a single OOD score for the baselines? For example, for Energy/MSP/MLS/ODIN, are you treating the m time intervals as classes, applying softmax over time bins or aggregate them by max/sum/mean or something else? HazardDev uses survival information, while your comparison methods are being forced onto logits in a way they were not really designed for. So I would not call this a "fair comparison". It's good as a baseline, but I would state that classification-style OOD scores are only weak baselines when naively applied to MTLR logits, not strong evidence that these methods are bad for survival OOD detection.

Non-critical but recommended:
* Fig. 2 is never referred to. Also, I would write out "ID" on first use in the main text (Figure 2).
Right now there are multiple times you define "ID" on page 2.
* Section 4.4: You state that "MTLR does not produce class confidence score", but this is partly true, since MTLR does produce a discrete survival/event-time distribution that we can use to define uncertainty-like scores -- for example, entropy, maximum probability over time bins, or cumulative incidence. So the motivation is okay, but slightly overstated.
* Section 6.2: It would make the comparison more fair if you included some survival-native baselines: for example, entropy of predicted event-time distribution, max event-time probability, or simple survival curve deviation. Right now, HazardDev is compared to outside-survival methods where we know their assumptions are mismatched, so the results are not that unsurprising.

[1] Fan S, Xu R, Dong Q, He Y, Chang C, Cui P. Stable Cox regression for survival analysis under distribution shifts. Nature Machine Intelligence. 2024;6:1525–1541. doi:10.1038/s42256-024-00932-5.

[2] Wen J, Yu C-N, Greiner R. Robust learning under uncertain test distributions: Relating covariate shift to model misspecification. Proceedings of the 31st International Conference on Machine Learning. PMLR. 2014;32(2):631–639.

[3] Pfisterer F, Harbron C, Jansen G, Xu T. Evaluating domain generalization for survival analysis in clinical studies. Proceedings of the Conference on Health, Inference, and Learning. PMLR. 2022;174:32–47.

[4] Zong Y, Ma Y, Van Keilegom I. Survival analysis under label shift. arXiv preprint arXiv:2506.21190. 2025.

[5] Shin J, Lee CH, Kang S. Weighted conformal prediction for survival analysis under covariate shift. arXiv preprint arXiv:2512.03738. 2025.

[6] Lillelund CM, Qi S, Greiner R, Pedersen CF. Position: Stop chasing the C-index when evaluating survival analysis models. ICML 2026. arXiv preprint arXiv:2506.02075. 2025.

---

> ### Author Response · Authors · 2026-06-28
>
> Thank you for the careful reading and the useful suggestions.
>
> Manuscript revisions: We revised the related work to better cover survival distribution shift and conformal survival analysis, and added a more explicit description of the benchmark partitioning and training protocol. We now specify that we report a Harrell-style C-index, describe how the MTLR output is converted into a risk score, and report endpoint- and split-specific censoring rates. In addition, we added IBS as a distribution-aware survival metric and bootstrap confidence intervals for AUROC/AUPRC in the appendix.
>
> 1. Survival-specific related work. We added a subsection covering robust and stable Cox, domain generalization, label shift, and conformal prediction under covariate shift.
> 2. Table 2 caption and training sizes. We rewrote the caption to define the Train, ID, and OOD columns, as well as the ID/OOD ranges. The original training set used a stricter inclusion rule. For the training set of the slice thickness, we additionally required the other acquisition parameters to remain within their ID ranges, with exposure time at the ID value of 1000 ms and pixel spacing below 1.045 mm. This was done to avoid simultaneous shifts in other parameters alongside the controlled parameter. In the revision, we relax this filter and follow the standard split protocol. We now fix the training set to 1,462 ID cases for all four tasks, which is the largest size consistently available across splits after reserving the ID and OOD test sets. The remaining cases are held out from training.
> 3. Eq. 4: signed shift, not curve distance. We recognize that the original name “deviation” may have been misleading. The signed sum is intentional, so we renamed it to HazRes and revised its motivation to make this clearer. Our method is motivated by the idea that, for a survival model fitted on the ID training distribution, atypical inputs may activate risk-related patterns and produce higher hazard predictions. OOD samples with higher predicted hazard tend to receive positive HazRes scores, whereas ID samples with lower predicted hazard tend to receive negative scores. Therefore, HazRes keeps the direction of the shift relative to the training hazard curve, which helps detect OOD samples. In contrast, L1 and L2 distances ignore the sign of the shift and treat both positive and negative differences as OOD evidence. This may reduce the separation between ID and OOD scores, making OOD detection harder. We also corrected Eq. 4 so the test hazard is explicitly predicted from the input.
> 4. C-index measures only ranking. We agree that the C-index does not fully assess the accuracy of the predicted event-time distribution. However, it is still an appropriate metric for comparing survival prediction performance between the ID and OOD test sets. We also added IBS as a distribution-aware metric in Appendix C, whose results are consistent with the C-index findings.
> 5. C-index specification and censoring. We now specify Harrell's C-index, derive the risk score by summing the cumulative hazard over time from the MTLR distribution, report endpoint- and split-specific censoring rates in Appendix B, and state that the main C-index does not use IPCW. IBS is computed with IPCW to handle censored data.
> 6. Hyperparameters and model selection. We hold out one fifth of the training set as a validation set for early stopping and checkpoint selection. The ID and OOD test sets are never used for training, tuning, or selection.
> 7. Table 3 pooling and uncertainty. Thank you for pointing this out. We do test the ID and OOD samples together, and we now state this clearly. We added 95% bootstrap confidence intervals for AUROC and AUPRC in Appendix D.
> 8. Section 6.2 baseline adaptation. We now specify that the interval-wise MTLR logits are treated as class-logits and each post-hoc score is applied per its original definition. We tightened the claim: these are baselines from naive adaptation, not evidence that such methods are unsuitable for survival OOD detection, and we no longer call it a "fair comparison".
> 9. Survival-native baselines. We added hazard-based baselines in Appendix E, including HazardPos and unsigned Hazard-L1 and Hazard-L2. Our method stays competitive across tasks and shifts.
> 10. Minor fixes. We now reference Figure 2, write out ID at first use, and remove repeated definitions. We have also revised the wording in Section 4.4 to be more measured.
> 12. Broader Impact. Thank you for the suggestion. We have added a Broader Impact Statement.

---

> > ### Comment · Reviewer_hu4u · 2026-06-29
> >
> > Thank you for the clarification and revision. I think this is good work and you have addressed all my concerns. However, one minor point remains: please provide the exact formula for the C-index risk score. The phrase "summing the cumulative hazard over time'' suggests $\sum_{i=1}^{m} H(t_i)$, which repeatedly weights early hazards, whereas you may instead mean summing the interval-specific hazards to obtain $H(t_m)$.

---

> > > ### Author Response · Authors · 2026-06-29
> > >
> > > Thank you for the positive review. We derive the risk score following the `torchmtlr` implementation. Specifically, the risk score is computed by summing the cumulative hazard,
> > > \begin{equation}
> > > r(\mathbf{x})=\sum_{j=1}^{m}H(t_j\mid\mathbf{x})
> > >             =\sum_{j=1}^{m}\sum_{i=1}^{j}h(t_i\mid\mathbf{x})
> > > \end{equation}
> > > This score is monotonically increasing with predicted hazard and monotonically
> > > decreasing with predicted survival probability, and therefore provides a scalar summary
> > > of the model-predicted survival curve. This is consistent with the
> > > MTLR-based framework used in prior work [1].
> > >
> > > [1] Chen, M., Wang, K. and Wang, J., 2024. Vision transformer-based multilabel survival prediction for oropharynx cancer after radiation therapy. International Journal of Radiation Oncology* Biology* Physics, 118(4), pp.1123-1134.

---

### Review · Reviewer_ZDUH · 2026-06-13

**Summary Of Contributions:**

The paper proposes a benchmark for out-of-distribution detection in survival prediction, a time-based regression task which aims to predict survival distributions from covariates. The benchmark is obtained by selecting a subset from another dataset, and the paper details the curation choices.

**Audience:**

Yes

**Audience Explanation:**

The benchmark curation process and the type of methods it aims at evaluating seem relevant to survival analysis.

**Broader Impact Concerns:**

No ethical concern, survival analysis is of great interest to learning in medical applications.

**Claims And Evidence:**

No

**Claims Explanation:**

There are several issues in the proposed method and analysis of the previous ones.

On baselines

The analysis of the baselines all consist to classification methods on the survival logits. Modern deep models may operate on learrned representations beyond the time step logits, as referenced in the paper (DeepSurv, deep-Cox). The negative result on time logits is valuable, but it should be contrasted with other approaches.

The authors also didn't compare their approach to the stream of literature on conformal survival analysis (Conformalized survival analysis with adaptive cutoffs, Gui et al 2023, Conformalized Survival Analysis, Candes et al 2021).

On the proposed method

HazardDev as the signed sum of differences of hazard values over all time steps. This indicates that hazard dev is an excessive hazard.
The authors justify HazardDev by arguing that OOD samples exhibit systematically higher hazard but this is an empirical observation on this dataset. Samples with an abnormally low hazard (one could imagine a defective scanner yielding a high rate of false positives) should still be considered out-of-distribution.

On the results
Figure 5 and Figure 10 make opposite conclusions in the captions -> OOD shows higher mean hazard in figure 5, ID show higher mean hazard in figure 10.

**Requested Changes:**

Figure 4 should be changed from a pie chart to a bar chart. Pie charts are overall a documented bad practice in data visualization.
See required clarifications above, baselines missing and justification of the proposed metric beyond the particular dataset.

---

> ### Author Response · Authors · 2026-06-28
>
> Thank you for the careful reading and the useful suggestions.
>
> Manuscript revisions:  We have revised the manuscript to better position CURE-OOD within the survival-specific distribution-shift literature, including robust survival learning, domain generalization, label shift, and conformalized survival analysis. We also replaced the pie chart in Fig. 4 with a bar chart, corrected the inconsistent hazard-curve caption, and added survival-native baselines in the appendix.
>
>
> 1. Baselines beyond time-step logits (representation-based detection). We agree that modern deep survival models operate on learned representations and that our logit-based negative result should be contrasted with representation-space approaches. In the revised benchmark we add NNGuide, a representation-space OOD detector that does not rely on the survival logits, as a reference baseline. The results show that our method achieves higher average AUROC and AUPRC than NNGuide across most tasks and acquisition shifts.
> 2. Conformal survival analysis. Thank you for pointing us to this literature. We now discuss it in the related work. Conformal survival methods aim to provide calibrated predictive sets, intervals, or survival-time bounds under censoring, and some variants further incorporate weighting for covariate shift. This objective is complementary to ours, as CURE-OOD evaluates whether a test sample is acquisition-shifted before its survival prediction is trusted. Since these methods address calibration rather than OOD detection, we discuss them in the related work rather than using them as baselines.
> 3. Motivation of the proposed score. We have revised the paper to better clarify the motivation of the proposed score. Our method is based on a one-sided risk-shift intuition. The assumption is that the survival model is well fitted on the ID training distribution. In this case, the model learns risk-related patterns from ID samples and produces hazard predictions aligned with the training population. Lower-risk ID patients often share more typical risk-related patterns, while higher-risk patients may show atypical features linked to worse outcomes. The model may therefore associate such atypical patterns with higher predicted hazard. When an input deviates from the ID distribution, it may activate similar risk-related patterns and lead to a higher predicted hazard. Our method uses this hazard shift as a survival-aware OOD signal. We agree that this score cannot cover all OOD cases. An OOD sample with abnormally low predicted hazard may not be detected by our method. We now state this limitation in the revised paper.
> 4. Figures 5 and 10. Thank you for pointing out this inconsistency. The contradiction came from a caption error and we have now corrected the caption.
> 5. Figure 4 (pie chart to bar chart). We have replaced the pie chart with a bar chart as suggested.